# Targeting the Annexin A1-FPR2/ALX pathway for host-directed therapy in dengue disease

Vivian Vasconcelos Costa[1†], Michelle A Sugimoto[2,3,4†], Josy Hubner[1], Caio S Bonilha[2,5], Celso Martins Queiroz-Junior[1], Marcela Helena Gonçalves-Pereira[2], Jianmin Chen[4], Thomas Gobbetti[4], Gisele Olinto Libanio Rodrigues[2], Jordana L Bambirra[6], Ingredy B Passos[1], Carla Elizabeth Machado Lopes[1], Thaiane P Moreira[6], Kennedy Bonjour[7], Rossana CN Melo[7], Milton AP Oliveira[8], Marcus Vinicius M Andrade[3], Lirlândia Pires Sousa[9], Danielle Gloria Souza[6], Helton da Costa Santiago[2], Mauro Perretti[4,10*‡], Mauro Martins Teixeira[2*‡]

[1]Department of Morphology, Institute of Biological Sciences, Universidade Federal de Minas Gerais, Belo Horizonte, Brazil; [2]Department of Biochemistry and Immunology, Institute of Biological Sciences, Universidade Federal de Minas Gerais, Belo Horizonte, Brazil; [3]School of Medicine, Universidade Federal de Minas Gerais, Belo Horizonte, Brazil; [4]William Harvey Research Institute, Barts and The London School of Medicine and Dentistry, Queen Mary University of London, London, United Kingdom; [5]Institute of Infection, Immunity and Inflammation, College of Medical, Veterinary and Life Sciences, University of Glasgow, Glasgow, United Kingdom; [6]Department of Microbiology, Institute of Biological Sciences, Universidade Federal de Minas Gerais, Belo Horizonte, Brazil; [7]Department of Biology, Institute of Biological Sciences, Federal University of Juiz de Fora, Juiz de Fora, Brazil; [8]Tropical Pathology and Public Health Institute, Universidade Federal de Goiás, Goiânia, Brazil; [9]Department of Clinical and Toxicological Analyses, School of Pharmacy, Universidade Federal de Minas Gerais, Belo Horizonte, Brazil; [10]Centre for Inflammation and Therapeutic Innovation, Queen Mary University of London, London, United Kingdom

*For correspondence:
m.perretti@qmul.ac.uk (MP);
mmtex.ufmg@gmail.com
(MMartinsT)

†These authors contributed
equally to this work
‡These authors also contributed
equally to this work

**Abstract** Host immune responses contribute to dengue's pathogenesis and severity, yet the possibility that failure in endogenous inflammation resolution pathways could characterise the disease has not been contemplated. The pro-resolving protein Annexin A1 (AnxA1) is known to counterbalance overexuberant inflammation and mast cell (MC) activation. We hypothesised that inadequate AnxA1 engagement underlies the cytokine storm and vascular pathologies associated with dengue disease. Levels of AnxA1 were examined in the plasma of dengue patients and infected mice. Immunocompetent, interferon (alpha and beta) receptor one knockout (KO), AnxA1 KO, and formyl peptide receptor 2 (FPR2) KO mice were infected with *dengue virus* (DENV) and treated with the AnxA1 mimetic peptide Ac$_{2-26}$ for analysis. In addition, the effect of Ac$_{2-26}$ on DENV-induced MC degranulation was assessed in vitro and in vivo. We observed that circulating levels of AnxA1 were reduced in dengue patients and DENV-infected mice. Whilst the absence of AnxA1 or its receptor FPR2 aggravated illness in infected mice, treatment with AnxA1 agonistic peptide attenuated disease manifestationsatteanuated the symptoms of the disease. Both clinical outcomes were attributed to modulation of DENV-mediated viral load-independent MC degranulation. We have thereby identified that altered levels of the pro-resolving mediator AnxA1 are of pathological relevance in DENV infection, suggesting FPR2/ALX agonists as a therapeutic target for dengue disease.

## Editor's evaluation

Costa, Sugimoto and colleagues report that the levels of anti-inflammatory Annexin A1 are reduced in dengue patients and in mice infected with dengue virus. They further show that absence of Annexin A1 or its FPR2/ALX receptor is associated with more severe disease, while treatment with an agonistic peptide had beneficial effects in mouse models. Their results suggest that strengthening Annexin A1 function may help to prevent severe dengue disease and agree with previous findings showing that treatment with corticosteroids that induce Annexin A1 may prevent the progression to severe illness.

## Introduction

Dengue is caused by one of four serotypes of dengue virus (DENV1-4) transmitted by *Aedes Aegypti* and *A. Albopictus* mosquitoes, affecting around 400 million people in 128 countries (*Bhatt et al., 2013*). Occasionally, dengue infection develops into a potentially lethal complication identified as severe dengue, typified by exacerbated systemic inflammation, vascular leakage, fluid accumulation, respiratory distress, severe bleeding, and/or organ impairment (*WHO, 2009*). No antiviral drug for dengue treatment is available, and the use of Dengvaxia (CYD-TDV), the first dengue vaccine approved by the US Food and Drug Administration, has its limitations, such as the increased risk for development of severe dengue in the immune populations (*Huang et al., 2021*; *WHO, 2018*; *The Lancet Infectious Diseases, 2018*). Thus, the combination of these factors points to dengue as a major unmet clinical problem in countries affected by this disease (*Shepard et al., 2016*; *Stanaway et al., 2016*).

The pathogenesis of severe dengue results from exacerbated host innate and adaptative immune responses to DENV. Among the target cells for DENV in humans, mast cells (MCs) lining blood vessels undergo dramatic cellular activation in response to DENV, despite their high resistance to infection (; *St John et al., 2013bSt John et al., 2011*). As essential regulators of vascular integrity, MC activation evoked by DENV triggers the production of inflammatory cytokines (cytokine storm) and vascular leakage, which ultimately result in hypovolemic shock in severe dengue (*St John et al., 2013b*; *Syenina et al., 2015St John et al., 2011*;). This is substantiated by a positive correlation between circulating chymase levels and disease severity recently observed in paediatric and adult patients (*Rathore et al., 2020*; *Tissera et al., 2017*). In line with the suggested role of uncontrolled immune responses in the pathogenesis of DENV infection, pharmacological suppression of inflammation (*Fu et al., 2014*; *Marques et al., 2015*; *Souza et al., 2009*), MC stabilisation (*St John et al., 2013bMorrison et al., 2017*), and inhibition of MC-derived protease (*Rathore et al., 2019*) have shown to be beneficial in experimental dengue. These studies provide proof-of-concept that host-directed therapies targeting excessive or misplaced inflammation may be a viable approach in treating severe dengue infection.

Pro-resolving mediators are cell signalling molecules synthesised in a strict temporal and spatial fashion to regulate the host response and prevent the excessive acute inflammatory reaction that damages the host (*Sugimoto et al., 2019*). The discovery of this active phase of inflammation has led to a new awareness of how a disease can emerge, including the concept that dysregulation or 'failure' in pro-resolving mechanisms might be involved in the pathogenesis of several chronic inflammatory disorders (*Eke Gungor et al., 2014*; *Fredman et al., 2016*; *Murri et al., 2013*; *Schett and Neurath, 2018*; *Sena et al., 2013*; *Thul et al., 2017*; *Vong et al., 2012*). The pro-resolving protein Annexin A1 (AnxA1) and its cognate receptor formyl peptide receptor 2 (FPR2), also known as FPR2/lipoxin A4 receptor (FPR2/ALX), bear anti-inflammatory properties in sterile settings (*Fredman et al., 2015*; *Galvão et al., 2017*; *Gimenes et al., 2015*; *Kusters et al., 2015*; *Leoni et al., 2015*; *Locatelli et al., 2014*), and exert a degree of protection in infectious settings, such as experimental tuberculosis (*Tzelepis et al., 2015*; *Vanessa et al., 2015*), sepsis (*Damazo et al., 2005*; *Gobbetti et al., 2014* *Spite et al., 2009*, pneumococcal pneumonia (*Machado et al., 2020*; *Tavares et al., 2016*), and influenza (*Schloer et al., 2019*). AnxA1 has been recently described to act as an endogenous modulator of MC degranulation in response to IgE/anti-IgE or compound 48/80 (*Parisi et al., 2019*; *Sinniah et al., 2019*; *Sinniah et al., 2016*; *Yazid et al., 2013*). Since AnxA1 is well known to counter regulate overexuberant pro-inflammatory events and MC activation, we have hypothesised that an imbalance between this anti-inflammatory/pro-resolving mediator and pro-inflammatory molecules could be operating during dengue infection. Whilst studies on bacterial infection consistently revealed the

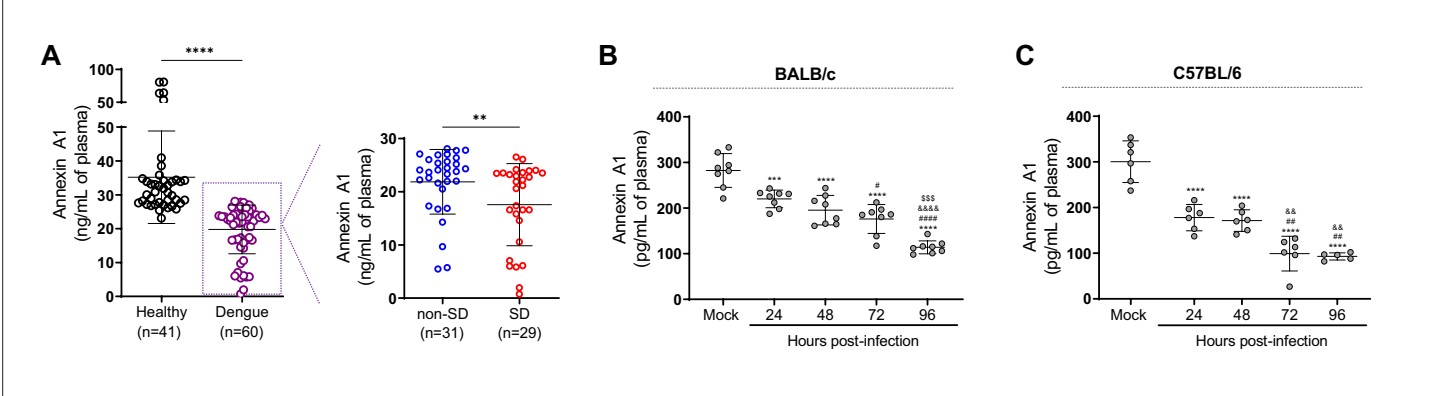

**Figure 1.** Annexin A1 (AnxA1) levels are reduced in dengue patients and mice infected with dengue virus (DENV). (**A**) AnxA1 plasma levels in healthy and dengue patients. The latter group was stratified into non-severe dengue (non-SD, outpatients) and severe dengue (SD, inpatients) individuals. Each circle represents an individual participant and horizontal bars represent mean values for AnxA1 (ng/mL of plasma), assayed by ELISA. ****p<0.0001, **p<0.01 by two-tailed Mann–Whitney test. (**B**) Five-week-old BALB/c (n = 8) or (**C**) C57BL/6 (n = 5–6) WT mice were intravenously injected with 1 × 10^6 PFU of DENV-2 and culled in the indicated time points for plasma collection. AnxA1 plasma levels analysed by ELISA are shown. ***p<0.001 and ****p<0.0001 versus mock-infected group; #p<0.05, ##p<0.01 and ####p<0.0001 versus 24-hr-infected group; &&p<0.01 and &&&&p<0.0001 versus 48-hr-infected group; $$$p<0.001 versus 72-hr-infected group (one-way ANOVA followed by Tukey's post hoc test).

The online version of this article includes the following source data for figure 1:

**Source data 1.** Raw data for *Figure 1A–C*.

ability of specialised pro-resolving mediators and AnxA1 (*Chiang et al., 2012*; *Decker et al., 2021*; *Gobbetti et al., 2014*; *Machado et al., 2020*; *Sekheri et al., 2020*; *Tzelepis et al., 2015*; *Vanessa et al., 2015*) to facilitate innate immune responses against the pathogen whilst reducing the harmful effects of inflammation, the translational potential of these results to viral infection was less clear.

In the present work, we have analysed the role of the pro-resolving AnxA1-FPR2/ALX pathway as a regulator of excessive inflammation observed in patients with the most severe forms of dengue infection. Our results suggest that failure to trigger this molecular pathway may contribute to disease severity in dengue infection and support the AnxA1-FPR2/ALX pathway as a potential target for host-directed therapy in human dengue disease.

## Results
### AnxA1 plasma levels are reduced in dengue patients

To ascertain how DENV infection impacts the expression dynamic of the pro-resolving molecule, AnxA1, in humans, we measured the AnxA1 protein level in the plasma of DENV-infected patients (*Figure 1A*). Dengue patients (n = 60) were grouped into non-severe dengue (SD) (dengue patients

**Table 1.** Demographics and laboratory characteristics of the study population from the control group, non-severe dengue (non-SD, outpatients), and severe dengue (SD, inpatients) groups during seasonal transmission 2013–2016.

| Characteristics and diagnosis of the study population | Control (n = 41) | Non-SD (n = 31) | SD (n = 29) |
|---|---|---|---|
| Age [a] | 30 (19–58) ** | 31 (17–65) * | 42 (19–76) |
| Gender (F) | 71% (29/41) | 55% (17/31) | 52% (15/29) |
| RT-PCR (n) | 0% (0/41) | 74% (23/31) | 38% (11/29) |
| ELISA IgM (n) | 0% (0/41) | 61% (19/31) | 100% (29/29) |
| Blood collection 1–5 days after symptom onset | - | 58% (18/31) | 45% (13/29) |
| Blood collection 6–12 days after symptom onset | - | 42% (13/31) | 55% (16/29) |

A geometric mean (min-max).

Blood collection: Fisher Test p>0.4.

Age: Kruskal–Wallis Test p<0.003. Gender: Fisher Test p>0.1.

that were treated at home as outpatients, n = 31) and SD (dengue inpatient cases that met WHO criteria for hospitalisation, n = 29) (*Gonçalves Pereira et al., 2020*). Demographics and laboratory characteristics are available in *Table 1*. Groups were comparable for sex distribution. In line with the association between older age in adults and an increased risk factor for progression to severe disease, our cohort was characterised by increased age in patients with SD (*Rowe et al., 2014*; *Sangkaew et al., 2021*). The proportion of patients showing secondary dengue infection was not significantly different between the non-SD and SD groups. All patients with positive polymerase chain reaction (PCR) reactions were infected with DENV-1, in line with a report that DENV-1 was the predominant serotype circulating in the city in which patients were recruited in the years of sample collection (*Gonçalves Pereira et al., 2020*). Outpatient evolution was confirmed by remote monitoring at the convalescent phase. Interestingly, we have identified that plasma levels of AnxA1 were reduced in dengue patients compared to healthy controls (*Figure 1A*). Stratification of groups according to disease severity at discharge showed that SD patients had discrete but significantly lower levels of AnxA1 compared to individuals with classic dengue (*Figure 1A*).

## DENV infection reduces AnxA1 plasma levels in mice

Since we identified reduced levels of AnxA1 in the plasma of dengue patients, we sought to investigate the role of this pro-resolving mediator in dengue's pathogenesis. To examine how the AnxA1 pro-resolving pathway operates in an immunologically intact system, we initially examined AnxA1 secretion over time in an immunocompetent animal model of DENV infection. Although wild-type (WT) mice are more resistant to infection than immunocompromised animals (*Shresta et al., 2004*); these animals were proven to be productively infected by DENV and are valid hosts to investigate the mechanisms underlying DENV-induced vascular dysfunction (*Chen et al., 2007*; *St John et al., 2013b*; *Syenina et al., 2015*). In both DENV-infected BALB/c (*Figure 1B*) and C57BL/6 mice (*Figure 1C*), there was a time-dependent decline in the concentration of plasma AnxA1 compared to mock-infected animals. Curiously, AnxA1 levels in human plasma were higher than in animal samples. Differences in AnxA1 plasma levels between both species have been previously noted (*Senchenkova et al., 2019*; *Xu et al., 2021*).

The decline in AnxA1 expression during DENV infection motivated investigation on the role of the AnxA1-FPR2/ALX pathway in dengue disease progression and severity.

## Disruption of the AnxA1-FPR2/ALX pathway aggravated illness and exacerbated inflammation in DENV-infected mice

We then conducted experiments in different animal strains to ascertain whether changes in AnxA1 expression could impact the dynamics of DENV infection (*Figure 2A*). As previously described (*St John et al., 2013b*), systemic infection of WT mice caused haematological and vascular changes consistent with the human disease, systemic MC activation, and inflammatory response (*Figure 2B–K*). Disease parameters were aggravated in AnxA1 KO animals compared with WT mice. In the absence of AnxA1, animals showed more severe and prolonged thrombocytopenia, haemoconcentration, and vascular permeability than WT mice (*Figure 2B–D*), indicating a protective role of AnxA1 in dengue disease. These findings were similar in FPR2-depleted animals (*Figure 2G–I*), suggesting that these effects are due to disruption of the AnxA1-FPR2/ALX pathway. Plasma levels of mast cell protease 1 (MCPT-1) and CCL2 were increased in either AnxA1 and FPR2 KO animals compared to their respective controls after infection (*Figure 2E, F, J and K*). While parameters, such as haematocrit and vascular leakage, returned to basal levels 72 hr after infection in WT animals, KO animals persisted with elevated levels, indicating a delay in resolving the host's haematological and immune response. Interestingly, AnxA1-depleted animals have preserved their ability to control virus spread, as splenic virus replication was similar to what was found in control animals (*Figure 2—figure supplement 1*).

## AnxA1 agonism attenuates DENV infection manifestations

Given the modulation of AnxA1 expression in DENV-infected mice and the protective role of the AnxA1-FRP2/ALX pathway in experimental disease, we next questioned whether exogenous administration of the AnxA1 mimetic peptide Ac$_{2-26}$ could attenuate dengue disease. Ac$_{2-26}$ binds to FPR2/ALX and mimics most of the whole protein's effects in experimental inflammation (*Sheikh and Solito, 2018*; *Sugimoto et al., 2016*). The peptide therapy in WT animals (*Figure 3A*) was effective after the

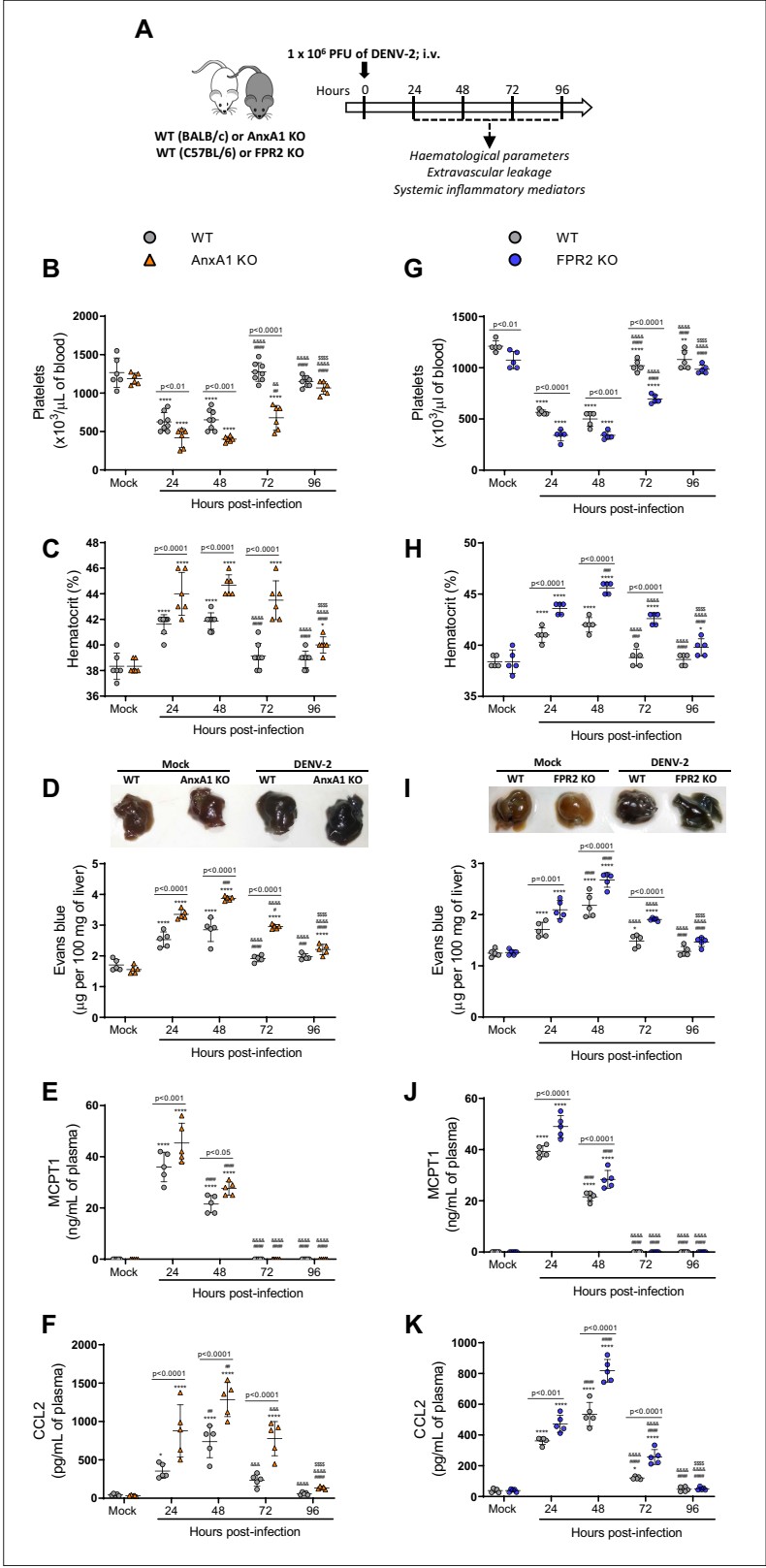

**Figure 2.** Mice are more susceptible to dengue virus (DENV-2) infection in the absence of Annexin A1 (AnxA1) or its receptor formyl peptide receptor 2 (FPR2). (**A**) Experimental design. (**B–F**) Five-week-old BALB/c WT and AnxA1 KO or (**G–K**) C57BL/6 and FPR2 KO mice were intravenously inoculated with $1 \times 10^6$ PFU DENV-2. Mice were culled in the indicated time points after infection and blood and tissue were collected for the following analysis: (**B,G**)

*Figure 2 continued on next page*

*Figure 2 continued*

platelet counts, shown as the number of platelets $\times 10^3$/μL of blood; (**C,H**) haematocrit levels, shown as % volume occupied by red blood cells; (**D,I**) vascular leakage assay with Evans blue dye; concentrations of (**E,J**) MCPT-1 and (**F,K**) CCL2 in plasma, quantified by ELISA and shown as quantity per mL of plasma. B–C, n = 6–8 animals per group; D–K, n = 5. Differences over time were compared by two-way ANOVA followed by Turkey's multiple comparison test: *p<0.05, **p<0.01, ***p<0.001, and ****p<0.0001 versus mock-infected group; #p<0.05, ##p<0.01, ###p<0.001, and ####p<0.0001 versus 24-hr-infected group; &p<0.05, &&p<0.01, &&&p<0.001, and &&&&p<0.0001 versus 48-hr-infected group; $p<0.05, $$p<0.01, $$$p<0.001, and $$$$p<0.0001 versus 72-hr-infected group. Differences between genotypes were compared by two-way ANOVA followed by Sidak's multiple comparison test, as indicated in the graphs. .

The online version of this article includes the following source data and figure supplement(s) for figure 2:

**Source data 1.** Raw data for *Figure 2B–K*.

**Figure supplement 1.** Dengue virus (DENV) replication is not altered by the absence of AnxA1.

first injection and protected against the major changes in haematological and immune markers as evident at 24 hr and 48 hr after DENV infection compared to mock-infected animals (*Figure 3B–F*). Thrombocytopenia (*Figure 3B*) and haemoconcentration (*Figure 3C*) were significantly milder in the group treated with Ac$_{2-26}$. DENV-2 infection provoked increased vascular permeability in both untreated and treated animals, but to a lesser extent when animals were administrated daily with AnxA1 mimetic peptide (*Figure 3D*). Of note, whilst increased haematocrit levels and vascular leakage were observed as early as 24-hr post-infection in vehicle-treated mice, treatment with AnxA1 mimetic peptide delayed the onset of both disease manifestations (*Figure 3B–C*). Treatment with Ac$_{2-26}$ also reduced the elevation in plasma MCPT-1 (*Figure 3E*) and CCL2 (*Figure 3F*) levels compared to the untreated group. Notably, the improvements in disease symptoms appeared to be independent of the splenic viral load (*Figure 3—figure supplement 1*).

When we applied the same treatment schedule to FPR2 KO mice infected with DENV-2, Ac$_{2-26}$ did not affect the disease parameters under observation (*Figure 3G–K*). Finally, the efficacy of Ac$_{2-26}$ in these settings allowed us to validate the data obtained with AnxA1 KO mice, as administration of AnxA1 peptide to animals deficient in AnxA1 rescued the phenotype showed by this transgenic colony in dengue infection, bringing values of haematological and immune parameters in line with those measured in untreated mice (*Figure 3—figure supplement 2*). Taken together, these results reveal the therapeutic potential of a pro-resolving peptide in the context of dengue, supporting the hypothesis that it could be operative also in settings with lower or absent AnxA1.

## Protective effects of Ac$_{2-26}$ are independent of the control of viral loads and virus dissemination

To establish whether Ac$_{2-26}$ treatment could have a therapeutic benefit after the infection is established and if it affects viral loads, we applied a different experimental system (*Figure 4A*), using mice bearing a null mutation for the interferon (alpha and beta) receptor 1 (IFNα/β KO; A129 mice). These animals are highly susceptible to DENV infection and present severe macroscopic and microscopic alterations (*Costa et al., 2013*; *Costa et al., 2014*; *Lam et al., 2017*; *Shresta et al., 2004*). As seen for BALB/c and C57BL/6 strains, DENV-infected A129 mice showed reduced AnxA1 plasma levels over the course of infection, compared with mock-infected animals (*Figure 4B*). While untreated A129 mice lost ~10% of their body weight from day 2 post-infection until the last time point analysed, treatment with Ac$_{2-26}$ substantially delayed weight loss onset (*Figure 4C*). The AnxA1 mimetic significantly reduced thrombocytopenia and haemoconcentration in response to DENV infection (*Figure 4D and E*) and displayed efficacy on controlling innate immunity mediators, with more significant effects in spleen values rather than plasma levels for CCL5 and IL-6 (*Figure 4F and G*). As observed in immunocompetent mice, Ac$_{2-26}$ attenuated the systemic release of MCPT-1 induced by DENV-2 (*Figure 4H*), although it did not affect systemic levels of CCL2 (*Figure 4I*). Given the characteristic of early DENV infection in our model, it is expected that the blood level of these markers would possibly not be altered by the analysed time point (5 d.p.i.). Infection of A129 mice induced a degree of liver damage, monitored by elevated plasma levels of alanine aminotransferase (ALT) and high-histopathological score (*Figure 4J–K*). Remarkably, treatment with Ac$_{2-26}$ attenuated liver injury caused by DENV, as indicated by reduced histopathological score (*Figure 4J and K*) and ALT transaminase levels (*Figure 4L*). Ac$_{2-26}$

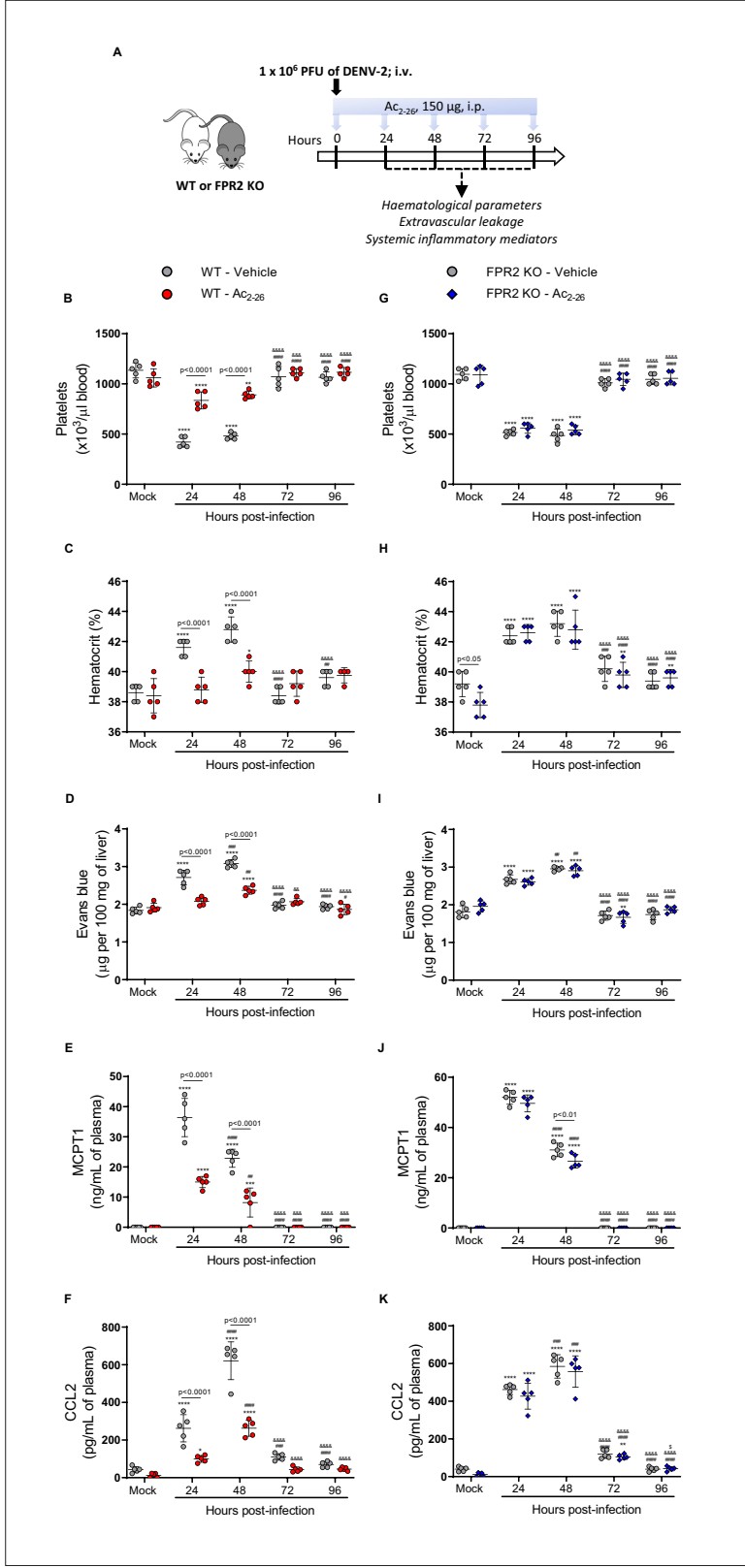

**Figure 3.** Annexin A1 mimetic peptide Ac$_{2-26}$ improves DENV-induced manifestations in wild-type (WT) mice and does not affect animals lacking its receptor formyl peptide receptor 2 (FPR). (**A**) Experimental design. Five-week-old BALB/c WT (**B–F**) and FPR2 KO (**G–K**) mice were intravenously inoculated with 1 × 10$^6$ PFU DENV-2. Mice were treated or not with 150 µg Ac$_{2-26}$ at the time of infection and daily thereafter by the intraperitoneal route. Mice were

*Figure 3 continued on next page*

*Figure 3 continued*

culled in the indicated time points after infection, and blood and tissue were collected for the following analysis: (**B,G**) platelet counts, shown as the number of platelets $\times 10^3$ /µL of blood; (**C,H**) haematocrit levels, shown as % volume occupied by red blood cells; (**D,I**) vascular leakage assay with Evans blue dye; concentrations of (**E,J**) MCPT-1 and (**F,K**) CCL2 in plasma, quantified by ELISA and shown as quantity per mL of plasma. N = 5 animals per group, except for graph C where n = 4–5. Differences over time were compared by two-way ANOVA followed by Turkey's multiple comparison test: *$p<0.05$, **$p<0.01$, ***$p<0.001$, and ****$p<0.0001$ versus mock-infected group; #$p<0.05$, ##$p<0.01$, ###$p<0.001$, and ####$p<0.0001$ versus 24-hr-infected group; &$p<0.05$, &&$p<0.01$, &&&$p<0.001$, and &&&&$p<0.0001$ versus 48-hr-infected group; $$p<0.05$, $$$p<0.01$, $$$$p<0.001$, and $$$$$p<0.0001$ versus 72-hr-infected group. Differences between genotypes were compared by two-way ANOVA followed by Sidak's multiple comparison test, as indicated in the graphs.

The online version of this article includes the following source data and figure supplement(s) for figure 3:

**Source data 1.** Raw data for *Figure 3B–K*.

**Figure supplement 1.** Dengue virus (DENV) replication is not altered by treatment with Annexin A1 mimetic peptide $Ac_{2-26}$.

**Figure supplement 2.** $Ac_{2-26}$ peptide improve dengue virus (DENV)-induced manifestations in Annexin A1 (AnxA1) KO mice.

---

alone (in the absence of infection) did not cause any notable adverse effects, impacted inflammatory and haematological parameters, or caused liver damage (*Figure 4C–I and L*). In this animal strain and using this protocol, we could test the effects of $Ac_{2-26}$ following infection with other DENV serotypes and observed a significant reduction in haematological alterations, liver damage, and IL-6 production induced by either DENV-1, DENV-3, or DENV-4 (*Figure 5*).

Finally, we investigated the potential impact of the $Ac_{2-26}$ peptide on viral loads and virus dissemination. A129 mice showed systemic viral burden on day five after DENV1-4 inoculation, with detectable viremia and viral load in spleen and liver (*Figures 4M, 5F and G*). Treatment with $Ac_{2-26}$ did not affect systemic viral burden, as untreated and treated mice showed similar viremia and viral loads. Treatment with $Ac_{2-26}$ caused only a slight reduction in viral loads in the spleen of mice infected with DENV-2 (*Figure 4M*) and in the plasma of animals inoculated with DENV-3 (*Figure 5F*). These data indicate that the AnxA1 mimetic positively impacts this severe dengue model, exerting little or no control on virus dissemination and viral loads, thus genuinely regulating the host response. Moreover, we show that the efficacy of the AnxA1 peptide is not restricted to a single virus serotype.

## $Ac_{2-26}$ prevents mast cell degranulation evoked by DENV

There is compelling evidence that mast cell (MC) degranulation contributes to DENV-induced vascular leakage and disease severity (*St John et al., 2013b*; *Syenina et al., 2015*; *Tissera et al., 2017*). In line with this, we report increased MCPT-1 levels in the plasma of DENV-infected mice, which is enhanced in the absence of functional AnxA1/FPR2 and is counterbalanced by AnxA1 peptide treatment (*Figure 2E and J*; *Figure 3E, J* and *Figure 3—figure supplement 2E*). We hypothesised that the AnxA1 peptide could, at least in part, exert its protective effects by preventing MC degranulation. To test this hypothesis, we took advantage of in vivo and in vitro systems. We first analysed MCs from hind paw histological sections of WT mice pretreated with vehicle or AnxA1 mimetic peptide and infected with DENV (*Figure 6A*). Both animals treated locally or systemically had decreased MC degranulation in comparison with control mice. After verifying that systemic administration of $Ac_{2-26}$ prevents mast cell degranulation in vivo, we subjected AnxA1 KO and FPR2 KO mice to the same challenge and treatment schedule (*Figure 6B and C*). We verified that $Ac_{2-26}$ successfully reduces mast cell degranulation in AnxA1 KO mice (*Figure 6B*) but failed to rescue DENV-induced degranulation in FPR2 KO mice (*Figure 6C*), confirming the dependence of this receptor for its therapeutic actions. Notably, the absence of FPR2 significantly increased mast cell degranulation in response to the local challenge with DENV compared to the WT counterparts (*Figure 6C*).

To confirm a direct effect of the peptide in MC function, we cultured bone marrow-derived mast cells (BMMCs) and added $Ac_{2-26}$ before challenging cells with DENV-2. We then assessed BMMC degranulation in response to DENV-2 by a standard β-hexosaminidase assay. $Ac_{2-26}$ inhibited β-hexosaminidase release evoked by DENV-2 in a concentration-dependent manner, with ~40% reduction in release at

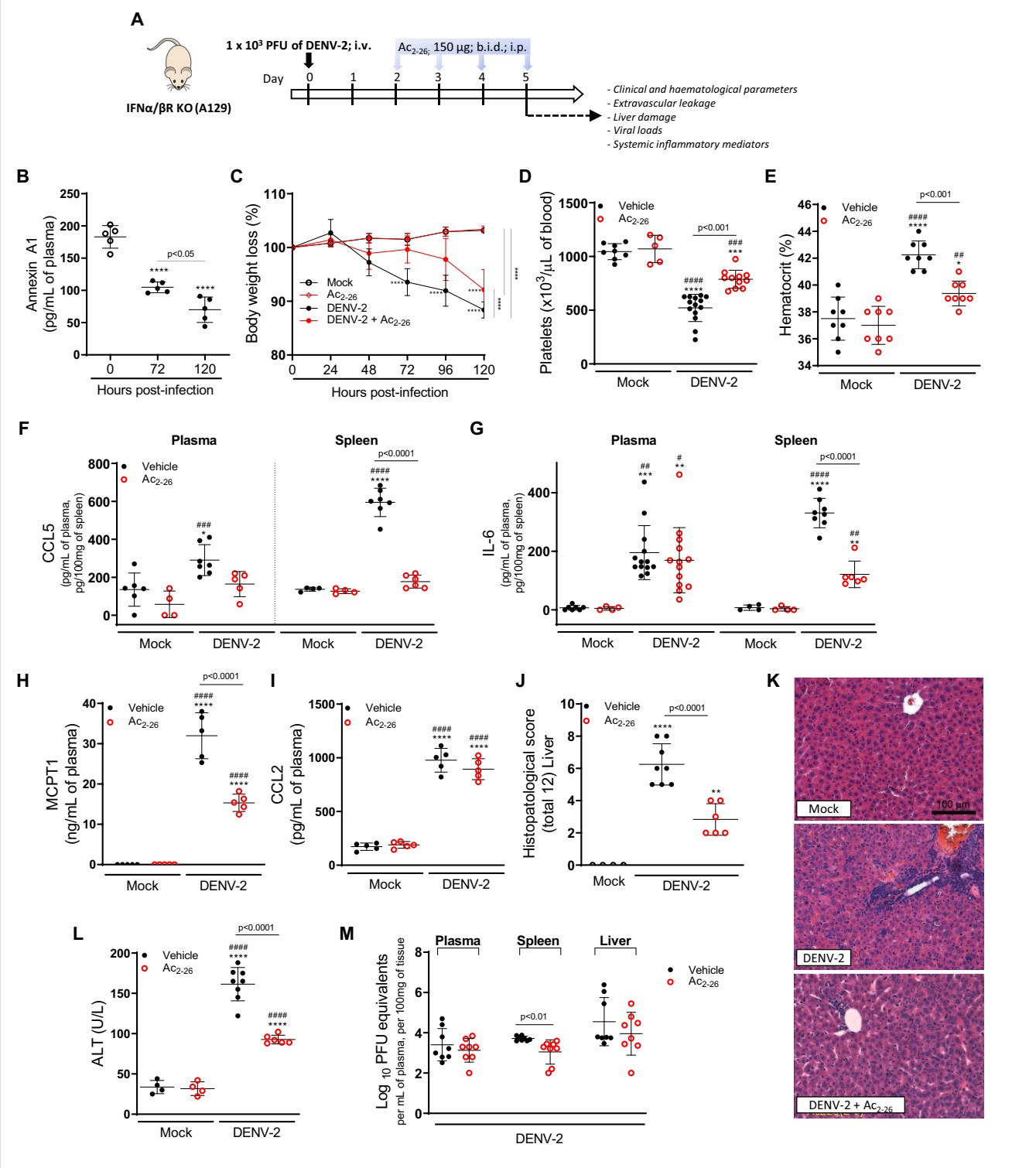

**Figure 4.** The protective effect of Ac_{2-26} administration in dengue virus (DENV)-infected A129 mice is viral load-independent. (**A**) Experimental design. Eight-week-old A129 mice were mock-infected or inoculated with $1 \times 10^3$ PFU of DENV-2 by the intravenous route. From day 2, mice, were treated or not twice a day with 150 µg of Ac_{2-26} by the intraperitoneal route. (**B**) Mice were culled in the indicated time points after infection, and plasma was collected for AnxA1 quantification by ELISA (n = 5). (**C**) Bodyweight loss was assessed in the indicated time points and expressed as a percentage of initial body weight. Mock (open white circles), Ac_{2-26} (open red circles), and DENV-2-infected mice treated with vehicle (black closed circles) or Ac_{2-26} (red closed circles); n = 4–8. Five days after infection, animals were culled, and blood and tissue collected for the following analysis: (**D**) platelet counts, shown as the number of platelets $\times 10^3$ /µL of blood (n = 5–14); (**E**) haematocrit levels, shown as % volume occupied by red blood cells (n = 8);

*Figure 4 continued on next page*

*Figure 4 continued*

concentrations of (**F**) CCL5 and (**G**) IL-6 in plasma and spleen of mock and DENV-infected mice, treated or not with Ac$_{2-26}$. Plasma Concentrations of (**H**) MCPT-1 and (**I**) CCL-2 in plasma of mock and DENV-infected mice, treated or not with Ac$_{2-26}$. Cytokines and chemokines were assessed by ELISA and are shown as quantity per mL of plasma or per 100 mg of the spleen (CCL5, n = 4–7; IL-6, n = 4–13; MCPT-1, n = 5; CCL-2, n = 5). (**J,K**) Liver of control and DENV-2-infected mice, treated or not with the AnxA1 peptide, were collected, formalin-fixed, and processed into paraffin sections. (**J**) Histopathological scores and (**K**) representative images of liver sections stained with haematoxylin and eosin. Scale Bar, 100 μm. (**L**) Plasma alanine aminotransferase activity represented as units/L (H–J, n = 4–8). (**M**) Viral loads recovered from plasma, spleen, and liver of infected mice treated or not with Ac$_{2-26}$, examined by plaque assay in Vero cells. Results are shown as the log of PFU/mL of plasma or as the log of PFU/mg of spleen and liver (n = 8). All results are expressed as mean (horizontal bars) ± SD. In C, differences over time and between treatments were compared by one-way ANOVA followed by Tukey's multiple comparisons test: ****p<0.0001 versus mock-infected animals or comparing the different groups, as indicated in the graph. In B,D–L, data were analysed by one-way ANOVA followed by Tukey's multiple comparisons test: *p<0.05, **p<0.01, ***p<0.001, and ****p<0.0001 versus mock-infected group; #p<0.05, ##p<0.01, ###p<0.001, and ####p<0.0001 versus mock-infected group treated with Ac$_{2-26}$. In M, statistical analyses were performed by unpaired Student's *t*-tests for each organ.

The online version of this article includes the following source data for figure 4:

**Source data 1.** Raw data for *Figure 4B–J and L–M*.

100 μM Ac$_{2-26}$ (*Figure 6D*). To confirm this observation, we performed a transmission electron microscopy (TEM) assay.

Ultrastructural analysis of mock-stimulated cells revealed morphological features of MCs in the process of maturation with cytoplasmic granules accumulating focal, rounded aggregates of electron-dense material (*Figure 6G*; *Combs, 1971*; *Dvorak et al., 1982*). Ultrastructural evidence of

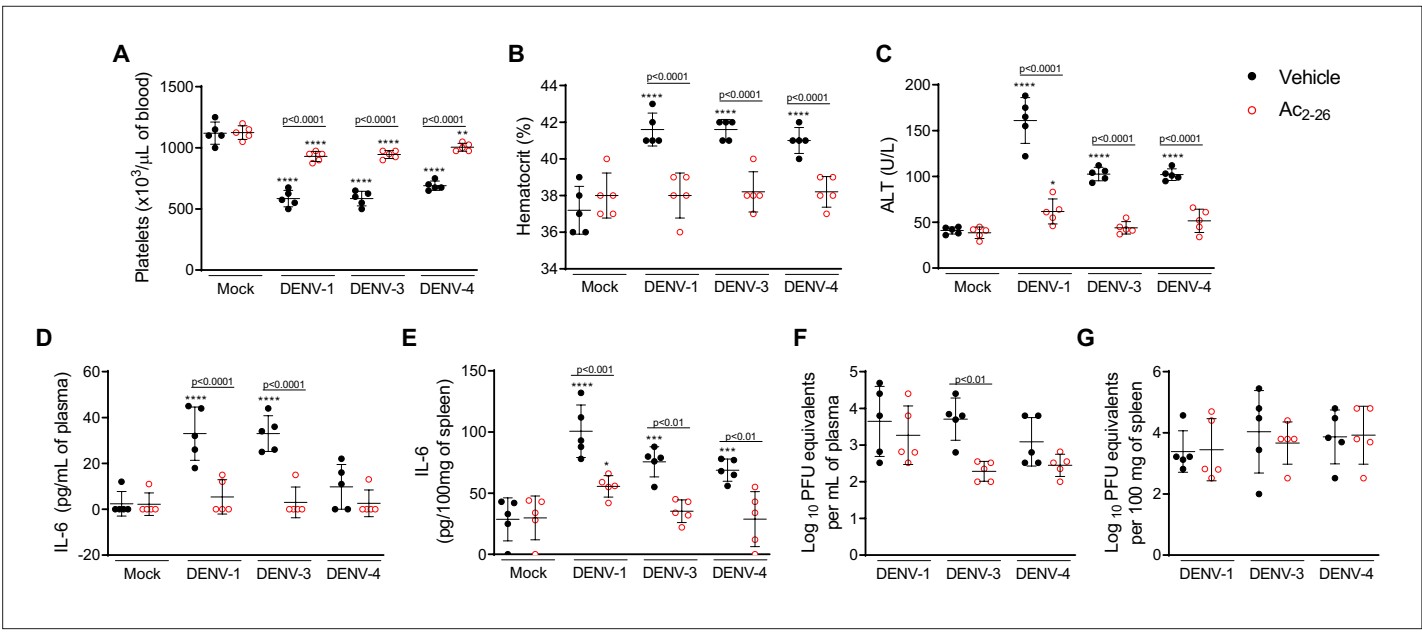

**Figure 5.** Treatment with Ac$_{2-26}$ ameliorates disease induced by different dengue virus (DENV) serotypes, without significantly impacting viral loads. Eight-week-old A129 mice were mock-infected or inoculated with 4 × 10⁴ PFU of DENV-1, 1 × 10³ PFU of DENV-3 or 1 × 10⁴ PFU of DENV-4 by the intravenous route (n = 5). From day 2, mice were treated with vehicle (black closed circles) or 150 μg of Ac$_{2-26}$, intraperitoneal route twice a day (open red circles). Five days after infection, animals were culled, and blood and tissue were collected for the following analysis: (**A**) Platelet counts, shown as the number of platelets × 10³ /μL of blood. (**B**) Haematocrit levels, shown as % volume occupied by red blood cells. (**C**) Plasma alanine aminotransferase activity represented as units/L. Concentrations of L-6 in (**D**) plasma and (**E**) spleen of mock- and DENV-infected mice, treated or not with Ac$_{2-26}$, assessed by ELISA. Concentrations are shown as pg/mL of plasma or as pg/100 mg of the spleen. Viral loads recovered from (**F**) plasma and (**G**) spleen of mice infected with the three serotypes of DENV and treated or not with Ac$_{2-26}$, evaluated by plaque assay in Vero cells. Results are shown as the log of PFU/mL of plasma or as the log of PFU/mg of spleen and liver. In A–E, data were analysed by two-way ANOVA followed by Dunnett's (*p<0.05, **p<0.01, ***p<0.001, and ****p<0.0001 versus mock-infected group) or Šídák's (statistical differences between infected mice treated with vehicle or Ac$_{2-26}$, as indicated in the graphs) multiple comparison test. In F–G, statistical analysis was performed by two-way ANOVA followed by Šídák's multiple comparison test, and differences between animals treated with vehicle or Ac$_{2-26}$ are indicated in the graphs. Horizontal bars represent mean values.

The online version of this article includes the following source data for figure 5:

**Source data 1.** Raw data for *Figure 5A–G*.

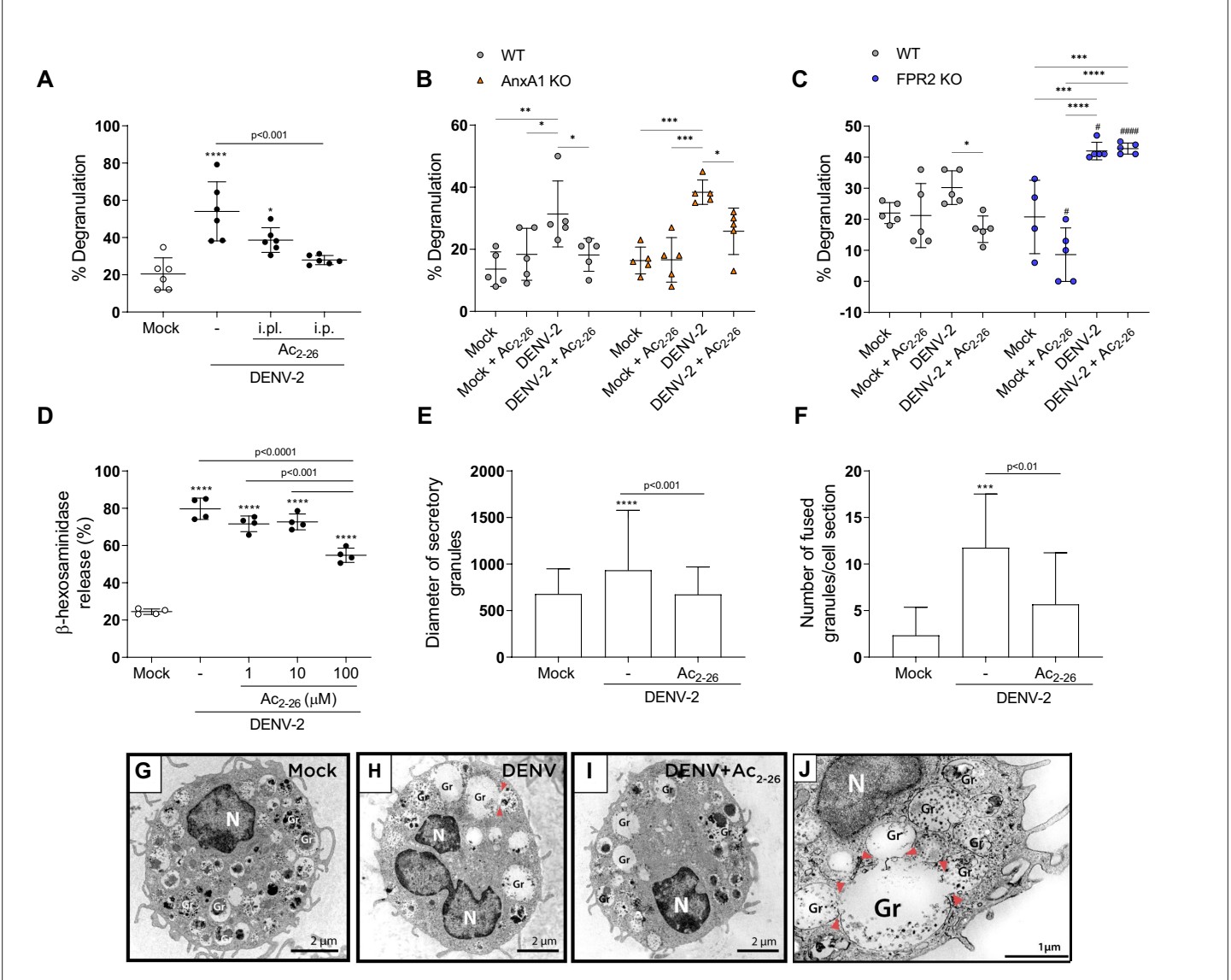

**Figure 6.** Ac$_{2-26}$ diminishes mast cell degranulation induced by dengue virus (DENV) both in vivo and in vitro. (**A**) WT BALB/c mice were treated with Ac$_{2-26}$ or vehicle via footpad injections (i.pl.; 100 μg; 10 min prior to infection) or systemically (intraperitoneal (i.p.) route; 150 μg; 1 hr prior to infection) and infected with $1 \times 10^6$ PFU of DENV-2 via footpad injections (n = 6). Three hours later, mice were euthanised and had their hind paws removed to analyse MC degranulation. (**B**) WT BALB/c mice and AnxA1 KO mice or (**C**) WT C57BL/6 and FPR2 KO mice were pre-treated with Ac$_{2-26}$ (i.p; 150 μg) 1 hr before infection with $1 \times 10^6$ PFU of DENV-2 via footpad injections (n = 5 for all groups, except for FPR2 KO mock, n = 4). Three hours later, mice were euthanised and had their hind paws removed to analyse MC degranulation. In A–C, mock-infected animals received an i.pl. injection of vehicle (sterile saline). (**D**) β-hexosaminidase activity of mouse bone marrow-derived mast cells (BMMCs), pretreated or not with increasing concentrations of Ac$_{2-26}$ for 1 hr, and stimulated with DENV-2 (MOI of 1) for an extra 30 min (n = 4). The data show the percentage release of cellular β-hexosaminidase into the medium and represent two independent experiments. (**E–J**) Transmission electron microscopy of BMMC, pretreated or not with Ac$_{2-26}$ 100 μM for 1 hr and challenged with DENV-2 (MOI of 1) for an extra 30 min. Significant increases in (**E**) granule diameters and (**F**) number of fused granules occur after stimulation with DENV-2 compared to both mock-stimulated cells and DENV-stimulated cells treated with Ac$_{2-26}$ peptide. In E, bars represent the mean diameter ± SD of 372, 360, and 384 secretory granules analysed in the mock, DENV-2 and DENV-2+Ac$_{2-26}$ groups, respectively. In F, bars represent the mean number of fused granules ± SD analysed in 15 sections per group. (**G**) Mock-stimulated BMMC show maturing cytoplasmic granules accumulating electron-dense material. (**H**) Granule enlargement, emptying, and fusion are observed in response to DENV-2 infection. (**I**) Ac$_{2-26}$ peptide treatment reduces morphological features of secretion evoked by DENV infection. Scale bar, 2 μm. (**J**) Granule fusions (arrowhead) in untreated DENV-infected BMMC are seen in higher magnification (scale bar, 1 μm). A,D,F, statistical analysis was performed by one-way ANOVA followed by Tukey's multiple comparisons test: ***p<0.001 and ****p<0.0001 compared to mock-infected cells, or as depicted on the graphs. B,C, differences between genotypes were compared by two-way ANOVA followed by Sidak's multiple comparison test. #p<0.05, ####p<0.001 compared to WT animals receiveing the same treatment/challenge. Differences between treatments were compared by two-way ANOVA followed by Turkey's multiple comparison test as indicated

*Figure 6 continued on next page*

*Figure 6 continued*

in the graphs: *p<0.05, **p<0.01, ***p<0.001 and ****p<0.0001 .E, data analysed by Kruskal–Wallis followed by Dunn multiple comparison test. Gr, secretory granules; N, nucleus.

The online version of this article includes the following source data and figure supplement(s) for figure 6:

**Source data 1.** Raw data for *Figure 6A–F*.

**Figure supplement 1.** Effect of Ac$_{2\text{-}26}$ treatment in mast cell degranulation induced by dengue virus (DENV-2).

degranulation was identified in BMMC cultured with DENV (***Figure 6H and J***), in which cells showed enlarged cytoplasmic granules with reduced electron density and granule–granule fusion events – all morphological changes indicative of content release (***Carmo et al., 2016***). These features were consistently reduced when infected cells were pretreated with 100 μM Ac$_{2\text{-}26}$ peptide (***Figure 6I***). Quantification of the morphological changes in granules showed that Ac$_{2\text{-}26}$ prevented the increase in granule diameters caused by DENV (***Figure 6E***). In addition, quantitative TEM demonstrated a significant increase in the number of granule-granule fusions in response to DENV infection than the mock-stimulated group; a feature significantly attenuated upon Ac$_{2\text{-}26}$ pre-treatment (***Figure 6F***). Altogether, our data suggest that the AnxA1 mimetic peptide Ac$_{2\text{-}26}$ directly acts on MC diminishing its degranulation in response to DENV.

## Discussion

We present evidence that an inadequate engagement of the resolution circuit AnxA1-FPR2/ALX may contribute to dengue infection's pathogenesis with particular relevance for the cohort of patients affected by the most severe forms of the disease. By exploring resolution biology as a novel approach in dengue disease, both with respect to etiopathogenesis and pharmacological opportunity, we have identified that: (i) AnxA1 is downregulated in the plasma of dengue patients in comparison to healthy individuals and in DENV-infected mice in comparison to non-infected animals; (ii) depletion of the AnxA1-FPR2/ALX pathway aggravates clinical signs and enhances MC activation associated to DENV infection, indicating a nonredundant role for this resolution pathway in the pathogenesis of dengue disease; (iii) pharmacological treatment of mice with an FPR2/ALX agonistic peptide produced beneficial effects during DENV infection; (iv) AnxA1 mimetic peptide has direct inhibitory effects on MC degranulation induced by DENV while (v) it does not seem to control viral load and virus dissemination significantly.

In recent years, a new paradigm shift has emerged in our understanding of the pathogenesis of inflammatory diseases, which results from persistent and exacerbated pro-inflammatory signals and dysregulation or 'failure' in resolving mechanisms (***Perucci et al., 2017***) ***Eke Gungor et al., 2014***; ***Fredman et al., 2016***; ***Murri et al., 2013***; ***Schett and Neurath, 2018***; ***Sena et al., 2013***; ***Thul et al., 2017***; ***Vong et al., 2012***. The severity and lethality of several infectious diseases, like dengue and the flu, frequently arise from an excessive host response, characterised by an uncontrolled release of pro-inflammatory cytokines leading to over-exuberant immune activation (***Costa et al., 2013***; ***St John et al., 2013a***). Inflammation is physiologically balanced by resolution circuits, such as those centred on AnxA1 and lipid mediators (e.g. lipoxins, resolvins, protectins and maresins) that drive termination of inflammatory response yet helping the host to deal with the infective agent (***Basil and Levy, 2016***). In line with this, a growing body of evidence indicates that pro-resolving mediators are regulated during infection, contributing to the control and resolution of experimental and human infectious diseases (***Abdulnour et al., 2016***; ***Basil and Levy, 2016***; ***Chiang et al., 2012***; ***Frediani et al., 2014***; ***Oliveira et al., 2017***; ***Shirey et al., 2014***). On the other hand, dysregulation in the production and/or action of pro-resolving mediators might contribute to the pathogenesis of sterile (***Eke Gungor et al., 2014***; ***Fredman et al., 2016***; ***Murri et al., 2013***; ***Sena et al., 2013***; ***Thul et al., 2017***; ***Vong et al., 2012***) and infectious (***Cilloniz et al., 2010***; ***Colas et al., 2019***; ***Molás et al., 2020***; ***Morita et al., 2013***) diseases, including atherosclerosis, inflammatory bowel diseases and tuberculous meningitis. Modulation of AnxA1 in the context of viral infections has been less investigated, and clinical data are scarce (***Arora et al., 2016***; ***Molás et al., 2020***). Since recent evidence suggests a defective engagement of pro-resolving pathways during self-resolving infections (***Cilloniz et al., 2010***; ***Colas et al., 2019***; ***Molás et al., 2020***; ***Morita et al., 2013***), we queried whether such alterations could also be occurring in

dengue disease. Focus was given to the pro-resolving protein AnxA1 and its cognate receptor FPR2/ALX, a pathway that has been shown to exert a degree of protection in experimental tuberculosis (*Tzelepis et al., 2015*; *Vanessa et al., 2015*), sepsis (*Damazo et al., 2005*; *Gobbetti et al., 2014*), pneumococcal pneumonia (*Machado et al., 2020*; *Tavares et al., 2016*), and influenza (*Schloer et al., 2019*). Our data indicate that in addition to the already described early induction of an inflammatory response that may be harmful instead of protective, DENV infection is also characterised by sustained downregulation of molecular components of the AnxA1 pathway. The drop in circulating AnxA1 below basal levels observed in dengue patients in this study, especially in severe dengue, suggests that the pathogenesis and severity of the disease might be associated with a failure to engage mechanisms involved in endogenous anti-inflammatory signals and its resolution, such as those centred on AnxA1. Furthermore, we have identified a protective role for the AnxA1-FPR2/ALX pathway in DENV infection, as animals presented heightened signs of disease in the absence of either ligand or receptor, compared to WT mice. In line with integrated pro-resolving properties of resolution circuits, AnxA1 and FPR2 KO mice displayed diverse uncontrolled responses leading to increased disease severity. Markers of MC degranulation and systemic inflammation were elevated in transgenic mice compared to WT counterparts. Similarly, these genetically manipulated animals presented enhanced haematological alterations in our model of experimental dengue.

Infections are currently treated by drugs that target pathogens or inhibit their growth. In some infectious diseases, blocking inflammation pathways may be beneficial. While this approach might be successful in some infections, excessive inhibition of the immune response can also be associated with immunosuppression and increased mortality, as observed in septic patients (*Carlet et al., 2020*). Resolution pharmacology has been proposed as an alternative to balance the host response without hampering its ability to deal with the infection. Indeed, pro-resolving receptor agonists' exogenous administration has proven benefits in experimental infectious settings, including bacterial pneumonia (*Abdulnour et al., 2016*; *Machado et al., 2020*) and influenza (*Morita et al., 2013*; *Ramon et al., 2014*; *Schloer et al., 2019*). In the present work, we provide proof-of-principle that a pro-resolving tool, the Ac$_{2-26}$ peptide, can ameliorate clinical disease and reduce circulating inflammatory mediators in an FPR2/ALX-dependent manner. The AnxA1 peptide was effective even when administered from day 2 after the infection onset.

In our experimental models, viral load was seemingly unaffected by Ac$_{2-26}$ treatment, supporting a protective effect independent of viral infectivity. Despite the limitations of modelling dengue disease in immunocompetent animals, infection of AnxA1 KO and FPR2 KO animals (and their respective littermates BALB/c and C57BL/6 mice) with high inoculum of DENV-2 was applied to investigate the role of endogenous AnxA1/FPR2 in dengue disease. Although viremia is not detectable in immunocompetent mice, we noticed transient viral RNA detection in the spleen starting 24 hr after infection, peaking at 48 hr or 72 hr, followed by disappearance at 96 hr, in line with previous studies using the same model (*St John et al., 2013b*; *Syenina et al., 2015*). Neither genetic deletion of AnxA1 or treatment with Ac$_{2-26}$ affected this profile. To complement our studies with immunocompetent animals mice, we performed experiments in A129 mice since they are known to be very susceptible to flaviviruses infections (*de Freitas et al., 2019*; *Del Sarto et al., 2020*) and develop a more severe disease manifestation that emulates several features of severe dengue in humans (*Zandi et al., 2019*). The later is considered a gold standard viremia model to test potential antiviral therapies (*Meyts and Casanova, 2021*; *Williams et al., 2009*). Using this model, we aimed to evaluate if the treatment with peptide Ac$_{2-26}$ has any impact on viral loads as well as in disease and inflammatory parameters. Of note, we were able to address the quantification of the viral loads by the plaque assay technique, the gold standard methodology to evaluate viable viral particles in vivo. Therefore, in our study we show, by two different techniques, that is, quantification of viral RNA (immunocompetent mice) and quantification of viable particles (A129 mice), and in two different models (high and low DENV inoculums) that neither absence of AnxA1 nor treatment with peptide Ac$_{2-26}$ interferes with the host ability to deal with the infection, despite its beneficial pro-resolving effects.

DENV infection initiates pro-inflammatory responses aiming to control virus spread that ultimately contributes to the immunopathology of dengue. For instance, MC activation in response to DENV plays an essential role in DENV-induced vascular pathology, particularly concerning the plasma leakage that causes hypovolemic shock in severe dengue (*St John et al., 2013b*; *Syenina et al., 2015*). This is supported by a correlation between circulating chymase levels and disease severity in

humans (*Rathore et al., 2020*; *Tissera et al., 2017*). It has been recently described that AnxA1 acts as an endogenous modulator of MC degranulation in response to IgE/anti-IgE or compound 48/80, suggesting that this pro-resolving axis act as a brake in MC degranulation (*Sinniah et al., 2016*; *Yazid et al., 2013*). The present study confirmed that dengue disease is associated with increased plasma levels of MCPT-1 and identified enhanced secretion in animals lacking AnxA1 or FPR2/ALX. Herein, we have identified the ability of Ac$_{2-26}$ to reduce DENV-induced MC degranulation dose-dependently, a mechanism that might underpin the reduced MCPT-1 secretion and vascular dysfunction observed in AnxA1 peptide-treated animals. Together, in vivo and in vitro evidence suggest that Ac$_{2-26}$, at least in part, acts by attenuating MC degranulation evoked by DENV, protecting the host against vascular dysfunction associated with the disease. This mode of action points to this pathway as a relevant potential target for DENV infection treatment, as MCs are resistant to infection but play a key role in dengue pathogenesis (; *Rathore et al., 2019*; *Syenina et al., 2015*; *Tissera et al., 2017*).

Although no therapy is available for dengue disease beyond supportive care, recent studies have pointed towards a benefit of using corticoids in severe dengue with no significant adverse consequences or promotion of virus replication (*Bandara and Herath, 2018*; *Bandara and Herath, 2020*). Amongst the several genomic and non-genomic actions of glucocorticoids (GC), promotion of AnxA1 has shown to be relevant to the benefits afforded by steroids (*De Caterina et al., 1993*; *Solito et al., 2003*), as suggested by the loss of the therapeutic effect of GC on AnxA1 depleted animals (*Patel et al., 2012*; *Perretti and D'Acquisto, 2009*; *Vago et al., 2012*). Our study reveals a central role of the axis AnxA1-FPR2/ALX in the pathogenesis of dengue disease. We suggest that this critical anti-inflammatory/pro-resolving pathway is downregulated during dengue, contributing to unbalanced inflammation and the cytokine storm characteristic of the disease. Therefore, there is scope to hypothesise that the benefit of glucocorticoid in dengue disease could be partially attributed to promoting the glucocorticoid inducible Annexin A1-FPR2/ALX pathway. Further investigations should confirm AnxA1 as an underlying mechanism of GCs in dengue disease. The effectiveness of the treatment with AnxA1 peptide in our study suggests that GC therapy could be refined using a more targeted approach, particularly in infectious diseases where there is an underlying imbalance in the AnxA1 pathway. This opens an opportunity to explore small molecules targeting FPR2/ALX receptors, avoiding the spanning adverse consequences of glucocorticoids whilst targeting the defect in inflammation resolution operating in the disease settings. Further investigation on GC and GC-inducible AnxA1 is particularly relevant in the context of dengue disease, where the use of non-steroidal anti-inflammatory agents in dengue is discouraged, as they can increase the risk of bleeding (*WHO, 2009*).

Our results indicate that altered levels of the pro-resolving mediator AnxA1 are of pathological relevance in dengue disease. We show that inadequate engagement of resolution circuits contributes to the excessive inflammation observed in severe DENV infection. In addition, we provide evidence for the benefits of pharmacological therapy directed to modulating host immune responses in the absence of a direct antiviral effect. These findings point to a direction for future research on applying FPR2/ALX agonists as a therapeutic target for dengue disease. Since pathogenic immune responses are not exclusive to dengue but underlie the severity of several other viral diseases, including severe community-acquired pneumonia caused by viruses (*D'Elia et al., 2013*; *Perrone et al., 2008*) and COVID-19 (*Jose and Manuel, 2020*), our findings could be translated into different infectious settings, whereby targeting the AnxA1 pathway, with or without combination with antiviral drugs, holds promising therapeutic potential.

## Materials and methods
### Patient recruitment
Dengue outpatients were recruited at Primary Care Center Jardim Montanhês and Santo Ivo Hospital. Inpatients were recruited in Odilon Behrens Metropolitan Hospital and Santa Casa Hospital. Healthy volunteers, negative for anti-DENV IgG (PanBio-Alere), were recruited in the community (Belo Horizonte, Minas Gerais, Brazil). Recruitment was done between the years of 2013–2016. Blood samples were obtained from 41 healthy donors and 60 dengue patients. Dengue patients were categorised into SD and non-SD groups using the 2009 WHO guidelines (*WHO, 2009*) and the expert physician's judgment of disease severity (*Gonçalves Pereira et al., 2020*). All SD patients were in-hospitalised. Of the 60 dengue patients enrolled in this study, 29 were classified as SD, and 31 were non-SD

patients. Patients were included in this study if DENV infection was confirmed by dengue specific IgM capture ELISA (PanBio-Alere) and/or real-time reverse transcriptase-polymerase chain reaction (RT-PCR) conducted on all blood samples. Individuals with comorbidities such as diabetes, autoimmune diseases or obesity were excluded from this study. Samples collected from healthy volunteers and patients with confirmed DENV infection, and a clear discharge diagnosis of either SD or non-SD were selected for measuring plasma AnxA1 levels by ELISA.

## Mice

Female BALB/c and C57BL/6 mice were obtained from the Center of Bioterism of Universidade Federal de Minas Gerais (UFMG), Brazil. Annexin A1 KO (BALB/c background) (*Hannon et al., 2003*) and FPR2 KO (C57BL/6 background, transgenic strain lacking the genes *Fpr2* and *Fpr3*) (*Dufton et al., 2010*; *Dufton et al., 2011*) mice were bred and maintained at animal facilities of the Immunopharmacology Laboratory of UFMG. Some experiments were conducted in type I interferon receptor-deficient mice (A129), SV129 background, obtained from Bioterio de Matrizes da Universidade de Sao Paulo (USP), bred and maintained at animal facilities of the Immunopharmacology Laboratory of the UFMG. For experiments, 5-week-old WT mice and 8-week-old A129 mice were kept under specific pathogen-free conditions at a constant temperature (25°C) with free access to chow and water in a 12 hr light/dark cycle.

## Cell lines, monoclonal antibodies, and viruses

Vero CCL81 (code 0245) and Aedes albopictus C6/36 (code 0343) cells were obtained from Banco de Células do Rio de Janeiro (BCRJ) repository and cultured in RPMI 1640 medium (Cultilab) or L15 medium (Cultilab), respectively, supplemented with 10% of inactivated foetal bovine serum (Cultilab). Cells were routinely tested for mycoplasma and found negative. For in vivo and in vitro experiments, low passage human clinical isolates of DENV serotypes DENV-1 (EDEN 2402), DENV-2 (EDEN 3295), DENV-3 (EDEN 863), and DENV-4 (EDEN 2270) were propagated in *Aedes albopictus* C6/36 cells, and the supernatants of infected cells were harvested, filtered, concentrated, tittered by plaque assay, and stored at −80°C until use. All in vivo studies with the infectious viruses were performed in a biosafety level 2 facility of the Immunopharmacology lab of the Institute of Biological Sciences at UFMG.

## Infections and drug treatments

For DENV infection experiments, mice received an intravenous injection of $10^6$ PFU (BALB/c WT, C57BL/6 WT, AnxA1 KO, and FPR2 KO mice) (*St John et al., 2013b*; *Syenina et al., 2015*) or $10^3$ PFU (A129 mice) (*Costa et al., 2014*; *Lam et al., 2017*) of DENV. The i.v. route of infection was chosen aiming to bypass the immune responses responsible for rapid virus clearance in natural peripheral infection (*St John et al., 2013b*; *Syenina et al., 2015*). For treatment with the AnxA1 mimetic peptide (Ac$_{2-26}$), BALB/c WT, C57BL/6 WT, AnxA1 KO, and FPR2 KO mice were injected intra-peritoneally with Ac$_{2-26}$ (150 µg/animal; phosphate-buffered saline, PBS, as the vehicle) at the time of infection and daily after infection (*Damazo et al., 2006*; *Galvão et al., 2017*; *Perretti et al., 1993*; *Vago et al., 2012*). A129 mice were treated with Ac$_{2-26}$ (150 µg/animal; i.p.) twice a day from day 2 post-infection until sacrifice (day 5). Mice were randomly allocated into experimental groups using an MS Excel randomisation tool. All experiments were repeated at least twice.

## Blood parameters

Murine blood was obtained from the cava vein in heparin-containing syringes at the indicated time points under ketamine and xylazine anaesthesia (100 mg/kg and 10 mg/kg, respectively). The final concentration of heparin was 50 U/mL. Platelets were counted in a Neubauer chamber (*Costa et al., 2014*; *Costa et al., 2012*). Results are presented as the number of platelets per µL of blood. For haematocrit determination, blood was collected into heparinised capillary tubes (Perfecta) and centrifuged for 10 min in a haematocrit centrifuge (Fanem, São Paulo, Brazil) (*Costa et al., 2014*; *Costa et al., 2012*).

## Changes in vascular permeability

The extravasation of Evans blue dye into the liver was used as an index of increased vascular permeability, as previously described (*Costa et al., 2014*; *Saria and Lundberg, 1983*; *St John et al., 2013b*).

The amount of Evans blue in the tissue was obtained by comparing the extracted absorbance with a standard Evans blue curve read at 620 nm in a spectrophotometer plate reader. Results are presented as the amount of Evans blue per 100 mg of tissue.

## Cytokines, chemokines, and AnxA1 quantification

The concentrations of murine CCL2, CCL5, and IL-6 in plasma samples and tissue homogenates were measured using commercially available DuoSet ELISA Development Kits (R&D). The concentrations of the MC-specific product MCPT-1 in plasma samples were measured using a commercially available ELISA Ready-SET-Go! Kit (eBioscience). Human or murine AnxA1 ELISA kits (USCN Life Sciences Inc) were used to quantify plasma levels of AnxA1. All the immunoassays were performed according to manufacturers' instructions.

## Virus titration

A129 mice were assayed for viral titres in plasma, spleen, and liver. Blood samples were collected in heparinised tubes and centrifuged at 3000× $g$ for 15 min at room temperature. The plasma was collected and stored at –80°C until assayed. For virus recovery from the spleen and liver, the organs were collected aseptically in different time points and stored at –80°C until assayed. Tissue samples were weighed and grounded using a pestle and mortar and prepared as 10% (w/v) homogenates in RMPI 1640 medium without foetal bovine serum (FBS). Viral load in the supernatants of tissue homogenates and plasma samples were assessed by direct plaque assay using Vero cells as previously described (*Costa et al., 2012*). Results were measured as plaque-forming units (PFU) per 100 mg of tissue weight or per mL of plasma. The limit of detection of the assay was 100 PFU/g of tissue or per mL.

## Transaminase activity

The ALT activity was measured in individual plasma samples from A129 mice, using a commercially available colourimetric kit (Bioclin, Quibasa, Belo Horizonte, Brazil). Results are expressed as U/L of plasma.

## Histopathology

Liver samples from euthanised mice were obtained at the indicated time points. After that, samples were immediately fixed in 10% neutral-buffered formalin for 24 hr and embedded in paraffin. Tissue sections (4 μm thicknesses) were stained with hematoxylin and eosin (H&E) and evaluated under a microscope Axioskop 40 (Carl Zeiss, Göttingen, Germany) adapted to a digital camera (PowerShot A620, Canon, Tokyo, Japan). Histopathology score was performed as previously described (*Costa et al., 2012*), evaluating hepatocyte swelling, degeneration, necrosis, and haemorrhage added to a five-point score (0, absent; 1, minimal; 2, slight; 3, moderate; 4, marked; and 5, severe) in each analysis. A total of two sections for each animal were examined, and results were plotted as the mean of damage values in each mouse.

## MC in vivo degranulation

BALB/c mice received Ac$_{2-26}$ footpad (100 μg) or i.p. (150 μg) injections followed by inoculation with $10^6$ PFU of DENV-2 via footpad injections. Three hours later, mice were euthanised and had their hind paws removed and fixed in 10% neutral-buffered formalin for conventional histopathological processing. In a second group of experiments, BALB/c, C57BL/6, AnxA1 KO, and FPR2 KO animals were pre-treated for 1 hr with Ac$_{2-26}$ (150 μg, i.p.) or vehicle followed by inoculation with $10^6$ PFU of DENV-2 via footpad injections. Three hours later, mice were euthanised and had their hind paws removed and processed for histological analysis as above. According to their morphological characteristics, MCs were classified as degranulated or normal, as previously described (*St John et al., 2011*). Mock-infected controls received an i.pl. injection of 20 μL of sterile saline (0.9% sodium chloride).

## BMMC generation and in vitro degranulation

BMMCs were generated as previously described (*Andrade et al., 2011*; *Rådinger et al., 2015*). After four weeks, BMMCs were verified by flow cytometry to be >95% positive for the MC surface marker c-kit (CD117). To assess degranulation of BMMC in response to DENV, cells were mock-stimulated

or stimulated with DENV-2 at MOI of 1, for 30 min, at 37°C, in HEPES degranulation buffer (10 mM HEPES, 137 mM NaCl, 2.7 mM KCl, 0.4 mM sodium phosphate, 5.6 mM glucose, 1.8 mM calcium chloride, 1.3 mM magnesium sulphate, pH 7.4). For some groups, BMMCs were incubated with 1, 10, or 100 µM $Ac_{2-26}$ 1 hr before stimulation with DENV. Degranulation was determined from the release of the granule marker β-hexosaminidase, as previously described (*Andrade et al., 2011*; *Rådinger et al., 2015*). The experiment was repeated twice with BMMCs isolated from distinct animals and differentiated independently.

## Transmission electron microscopy

BMMCs pre-treated or not with $Ac_{2-26}$ peptide (100 µM, 1 hr) followed by mock-stimulation or stimulation with DENV-2 (MOI of 1 for 30 min, at 37°C) were observed by TEM. BMMCs obtained from three distinct animals were treated and stimulated individually. Unlike tissue-housed MCs, primary cell cultures of the MCs can generate cells in different maturation profiles, including early-stage and fully developed cells with typical secretory granules (*Combs, 1971*; *Combs, 1966*). Thus, cells from the biological triplicate were pooled in each group to reach a suitable mature cell number in our analysis.

Following treatment and stimulation, cells were immediately fixed in a mixture of freshly prepared aldehydes (1% paraformaldehyde and 1.25% glutaraldehyde) in 0.1 M phosphate buffer, pH 7.3, for 1 hr at room temperature and prepared for conventional TEM as before (*Melo et al., 2009*). Sections were mounted on uncoated 200-mesh copper grids (Ted Pella) before staining with lead citrate and examined using a transmission electron microscope (Tecnai Spirit G12; Thermo Fisher Scientific/FEI, Eindhoven, Netherlands) at 120 kV. A total of 203 electron micrographs were analysed to investigate morphological changes indicative of degranulation. In addition, 1116 secretory granules (n = 372, n = 360, and n = 384 in the mock, DENV-2 and DENV-2+$Ac_{2-26}$ groups, respectively) were counted in 45 electron micrographs showing the entire cell profile and the granule diameters, as well as the numbers of fused granules were quantified. Quantitative studies were performed using the Image J software (National Institutes of Health, Bethesda, MD, USA).

## RT-PCR

RNA from human plasma was obtained using PureLink Viral RNA/DNA Kits (Invitrogen). Amplifications were performed by qPCR using SuperScript III Platinum One-Step Quantitative RT-PCR System with ROX (Invitrogen) according to the manufacturer's instructions in the presence of primers and probes described previously (*Gonçalves Pereira et al., 2020*; *Hue et al., 2011*; *Supplementary file 1*). For quantifying the virus in the mouse, RNA was isolated with the RNeasy kit from Qiagen. Briefly, tissues were homogenised with the machine TissueLyserII in a small amount of the buffer using ceramic beads. Then the manufacturer's protocol to isolate the RNA was performed, followed by cDNA synthesis. For detecting DENV-2 in the spleen, reverse primer 5'- TTGCACCAACAGTCAATGTCTTCAGGTTC was used for cDNA synthesis, followed by RT-PCR using forward primer 5'-TCAATATGCTGAAACGCGCG AGAAACCG and reverse primer 5'-CGCCACAAGGGCCATGAACAG.

## Statistics

GraphPad Prism 9.1.2 was used to determine statistical significance. Determination of sample size was based on previous publications using the software G*Power 3.1 Software. The results were analysed using appropriate statistical tests, as indicated in figure legends. Data are represented as mean ± SD.

## Data sharing

All data have been included in the manuscript, and source data files have been provided for *Figures 1–6*.

## Acknowledgements

The authors would like to thank Ilma Marçal, Tania Colina, Frankcineia Assis, and Gilvania Santos for the technical support. The authors also would like to thank Prof Eng Eong Ooi from Duke NUS Medical School in Singapore for providing the DENV strains used in mouse experiments. This work received financial support from the Fapemig Hospedeiro em Dengue project, the Medical Research Council in the United Kingdom (Newton project MR/No17544/1), the National Institute of Science and Technology in Dengue and Host-microorganism Interaction (INCT dengue), a program funded by The

Brazilian National Science Council (CNPq, Brazil) and Minas Gerais Foundation for Science (FAPEMIG, Brazil). JC and MP are also funded by the William Harvey Research Foundation. This study was financed in part by the Coordenação de Aperfeiçoamento de Pessoal de Nível Superior (CAPES, Brazil) – Finance Code 001. MAS received a Newton International Fellowship from the Academy of Medical Sciences during the preparation of the manuscript. The authors also thank L'Oréal-UNESCO-ABC 'Para Mulheres na Ciência' prize granted to VVC.

## Additional information

### Competing interests

Mauro Perretti: is on the Scientific Advisory Board of ResoTher Pharma AS, which is interested in developing Annexin A1-derived peptides for cardiovascular settings. The other authors declare that no competing interests exist.

### Funding

| Funder | Grant reference number | Author |
|---|---|---|
| Fundação de Amparo à Pesquisa do Estado de Minas Gerais | Fapemig Hospedeiro em Dengue project | Mauro Martins Teixeira |
| Medical Research Council | MR/No17544/1 | Lirlândia Pires Sousa Helton da Costa Santiago Mauro Perretti Mauro Martins Teixeira Vivian Vasconcelos Costa |
| Conselho Nacional de Desenvolvimento Científico e Tecnológico | Instituto Nacional de Ciência e Tecnologia em Dengue | Lirlândia Pires Sousa Danielle Gloria Souza Mauro Martins Teixeira Vivian Vasconcelos Costa |
| Fundação de Amparo à Pesquisa do Estado de Minas Gerais | Instituto Nacional de Ciência e Tecnologia em Dengue | Lirlândia Pires Sousa Danielle Gloria Souza Mauro Martins Teixeira Vivian Vasconcelos Costa |
| Coordenação de Aperfeiçoamento de Pessoal de Nível Superior | Finance Code 001 | Michelle A Sugimoto Mauro Martins Teixeira |
| L'Oréal-UNESCO-ABC | Para Mulheres na Ciência" prize | Vivian Vasconcelos Costa |
| Coordenação de Aperfeiçoamento de Pessoal de Nível Superior | Pós-Doutorado/Capes (PNPD /CAPES) | Michelle A Sugimoto |

The funders had no role in study design, data collection and interpretation, or the decision to submit the work for publication.

### Author contributions

Vivian Vasconcelos Costa, Conceptualization, Data curation, Formal analysis, Funding acquisition, Investigation, Methodology, Project administration, Supervision, Validation, Visualization, Writing – original draft; Michelle A Sugimoto, Conceptualization, Data curation, Formal analysis, Investigation, Methodology, Project administration, Validation, Visualization, Writing – original draft; Josy Hubner, Thomas Gobbetti, Carla Elizabeth Machado Lopes, Thaiane P Moreira, Investigation; Caio S Bonilha, Visualization, Writing - review and editing; Celso Martins Queiroz-Junior, Data curation, Investigation, Visualization, Writing - review and editing; Marcela Helena Gonçalves-Pereira, Data curation, Investigation, Resources, Writing - review and editing; Jianmin Chen, Investigation, Writing - review and editing; Gisele Olinto Libanio Rodrigues, Ingredy B Passos, Rossana CN Melo, Data curation, Investigation, Methodology; Jordana L Bambirra, Investigation, Methodology; Kennedy Bonjour, Data curation, Formal analysis, Investigation, Methodology, Validation, Writing - review and editing; Milton

AP Oliveira, Resources; Marcus Vinicius M Andrade, Methodology, Resources, Supervision; Lirlândia Pires Sousa, Investigation, Methodology, Resources; Danielle Gloria Souza, Conceptualization, Investigation, Methodology, Supervision, Visualization; Helton da Costa Santiago, Conceptualization, Data curation, Funding acquisition, Project administration, Resources, Supervision, Visualization; Mauro Perretti, Mauro Martins Teixeira, Conceptualization, Funding acquisition, Methodology, Project administration, Resources, Supervision, Visualization, Writing – original draft

#### Author ORCIDs
Vivian Vasconcelos Costa (iD) http://orcid.org/0000-0002-0175-642X
Michelle A Sugimoto (iD) http://orcid.org/0000-0002-4527-6065
Lirlândia Pires Sousa (iD) http://orcid.org/0000-0002-1042-9762
Mauro Perretti (iD) http://orcid.org/0000-0003-2068-3331
Mauro Martins Teixeira (iD) http://orcid.org/0000-0002-6944-3008

#### Ethics
Human subjects: Human sample collection was approved by the Committee on Ethics in Research of the Universidade Federal de Minas Gerais (Protocol Numbers 24832513.4.0000.5149 and 66128617.6.0000.5149). All patients have provided signed informed consent.

This study was performed in strict accordance with the Brazilian Government's ethical and animal experiments regulations (Law 11794/2008) and the recommendations of the CONCEA (Conselho Nacional de Controle de Experimentação Animal) from Brazil. All animal experiments received prior approval from the Animal Ethics Committee (CEUA) of Universidade Federal de Minas Gerais (UFMG), Brazil (Protocol numbers: 169/2016 and 234/2019). All surgeries were performed under ketamine/xylazine anaesthesia, and every effort was made to minimise animal suffering.

#### Decision letter and Author response
Decision letter https://doi.org/10.7554/eLife.73853.sa1
Author response https://doi.org/10.7554/eLife.73853.sa2

---

## Additional files

#### Supplementary files
• Supplementary file 1. Oligo primers and probes used in clinical samples. Patients were included in this study if DENV infection was confirmed by dengue specific IgM capture ELISA and/or real-time reverse transcriptase-polymerase chain reaction (RT-PCR). RT-PCR was conducted in RNA purified from human plasma using the primers and probes depicted in this table.

• Transparent reporting form

#### Data availability
All data has been included in the manuscript, and source data files have been provided for Figures 1-6.

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
