## [Editor Report]

Costa, Sugimoto and colleagues report that the levels of anti-inflammatory Annexin A1 are reduced in dengue patients and in mice infected with dengue virus. They further show that absence of Annexin A1 or its FPR2/ALX receptor is associated with more severe disease, while treatment with an agonistic peptide had beneficial effects in mouse models. Their results suggest that strengthening Annexin A1 function may help to prevent severe dengue disease and agree with previous findings showing that treatment with corticosteroids that induce Annexin A1 may prevent the progression to severe illness.

---

## [Decision Letter]

**Decision letter after peer review:**

Thank you for submitting your work entitled "Targeting the Annexin A1-FPR2/ALX pathway for host-directed therapy in dengue disease" for further consideration by *eLife*. Your article has been reviewed by 3 peer reviewers, including Frank Hirchhoff as Reviewing Editor and Reviewer #1, and the evaluation has been evaluated by Jos Van der Meer as Senior Editor. The reviewers have opted to remain anonymous.

There are some significant remaining issues that need to be addressed, as outlined below.

Essential revisions:

1. The improvements in disease symptoms appeared to be independent of the viral load. Since AC2-26 had previously been shown to have anti-inflammatory properties these findings are, while intriguing in the context of disease, not necessarily novel. The authors should mention that corticosteroids have been previously examined for the treatment of dengue disease and may exert beneficial effects through the induction of Annexin A1 and ALXR expression. They should also discuss why FPR2/ALX agonists may be better than treatment with corticosteroids to suppress exaggerated inflammation.

2. It is well established that DENV does not replicate efficiently in mice, but infection can occur using mouse-adapted strains of DENV and/or immunocompromised mouse strains. Here, the authors use a combination of wt (Balb/C and C57BL/6) and mouse mutant lines (including AG129 mice which have been shown to support robust infection with mouse adapted DENV strains) and infect them with a very high dose of DENV2 isolate. There are very limited viremia data shown throughout the manuscript which makes is somewhat difficult to interpret some of the results. How much are the effects in all the non-AG129 mouse lines attributable to an acute response to the high inoculum (and no/very limited replication) vs actual infection?

3. In Figure 4, when using IFNα/βR-/- mice (AG129), IL-6 and CCL5 instead of MCPT-1 and CCL2 were used as markers of systemic inflammatory mediators, which is not consistent with the previous experiments done in immunocompetent mice. No results of MCPT-1 and CCL2 changes in AG129 mice was presented and the authors didn't explain why they changed the markers in this manuscript. Since MCPT-1 is a mast cell-specific product, if it is not decreased in AG129 mice treated with Ac2-26 after DENV infection, how would the authors explain the protective effects of Ac2-26 in AG129 mice in the context of DENV infection?

4. The authors showed AnxA1 memetic peptide Ac2-26 could suppress MC degranulation. However, the same experiments should also be done in parallel in AnxA1-/- and FPR2/ALX-/- mice and in BMMCs derived from them. If Ac2-26 can prevent MC degranulation, it is expected to see decreased degranulation in AnxA1 mice but not FPR2/ALX KO mice when infected with DENV, both in vivo and in BMMCs. Finally, the peptide concentrations in mice (150 µg/animal) and in vitro (100 µM) are pretty high. The authors should mention whether any side effects were observed.

*Reviewer #1 (Recommendations for the authors):*

Dengue virus infection is spread by mosquitos and typically causes high fever, headache, vomiting, muscle and joint pains, and a characteristic skin rash. Typically, these symptoms resolved within a week but in some cases, dengue disease becomes more severe and is leads to hemorrhagic fever or dengue shock syndrome. Currently, no specific treatment against dengue disease is available. In the present study, Costa and colleagues hypothesized that excessive inflammatory responses play a key role in severe dengue disease and that anti-inflammatory molecules may be reduced in infected individuals. Specifically, they show that the circulating levels of anti-inflammatory Annexin A1 are reduced in both dengue patients and in infected mice. They further report that lack of Annexin A1 or its receptor FPR2/ALX enhance the disease in infected mice, while treatment with Annexin A1 agonistic peptide had beneficial effects. They further propose that the clinical outcome is dependent on inflammation and mast cell activation rather than the levels of viral replication. The authors conclude that Annexin A1 plays a key role in the clinical outcome of dengue virus infection and that FPR2/ALX agonists may offer therapeutic perspectives. Strength of the present study are that the authors analyzed both patient as well as mouse sample. Studies in knockout animals, provide strong evidence that the attenuating effect of Annexin A1 on the inflammatory response indeed attenuates the severity of dengue disease. Additional strengths are the evidence that an agonistic peptide exerts protective effect. A limitation is that the concentrations of the peptide used to monitor its effects (100 µM) are pretty high. In addition, the treatment is not specific and the concept of utilizing anti-inflammatory agents is not novel. Previous studies suggested that corticosteroids may prevent the progression to severe dengue disease (Rathnasiri Bandaraa and Herathb, Heliyon, 2018). Corticosteroids exert their anti-inflammatory effects by the induction of Annexin A1 and ALXR expression. This link should be mentioned and its needs to be explained why FPR2/ALX agonists may be better than treatment with corticosteroids to suppress exaggerated inflammation.

1. The authors should mention that corticosteroids have been previously examined for the treatment of dengue disease and may exert beneficial effects through the induction of Annexin A1 and ALXR expression. They should also discuss why FPR2/ALX agonists may be better than treatment with corticosteroids to suppress exaggerated inflammation.

2. Some references seem flawed (e.g. line 53, The Lancet Infectious, 2018). Should be checked throughout.

3. Line 254: "Groups were comparable for sex distribution". Table 1 shows that the proportion of females was higher in the control group and that, on average, age was higher in patients with severe dengue disease. This should be mentioned in the text.

4. It would be interesting to know whether the levels of Annexin A1 in individuals with dengue infection (Figure 1A) are inversely correlated with the levels of inflammatory cytokines.

5. The peptide concentrations in mice (150 µg/animal) and in vitro (100 µM) are pretty high. The authors should mention whether any side effects were observed.

6. The Annexin A1 levels in patients were about 10-fold lower than in mice (Figure 1). This difference should be discussed.

7. The authors mention that treatment with the Ac2-26 peptide did not affect systemic viral burden. It seems somewhat surprising that attenuation of innate immune activation doesn't affect the levels of viral replication. Please discuss.

*Reviewer #2 (Recommendations for the authors):*

In this manuscript, Costa et al., investigated the role of AnxA1-FPR2/ALX pathway in disease severity of dengue virus (DENV) infection and explores the therapeutic benefits of AnxA1 memetic peptide in DENV infection using mouse model. The authors show that serum levels of pro-resolving protein AnxA1 are significantly decreased in patients with DENV infection and infected mice. They show that the absence of AnxA-FPR2/ALX pathway in mice lead to aggravated dengue disease parameters by measuring hematological parameters, vascular leakage, and inflammatory factors. They further demonstrate that administration of a AxnA1 memetic peptide, Ac2-26, could improve these disease parameters in WT, AnxA1-/-, but not mice lacking the AnxA1 receptor, FPR2/ALX. In IFNα/βR-/- mice which are more susceptible to DENV infection, treatment of Ac2-26 improved the clinical outcome along with the disease parameters. The authors finally expore the mechanism of Ac2-26's protective effects and showing that Ac2-26 prevents mast cell degranulation both in vitro and in vivo models.

Overall, this study provides new insights on the mechanisms that underlie severe dengue disease. Notably, the data presented here suggest that the AnxA1-FPR2/ALX pathway may be a suitable potential target for host-directed therapy and a biomarker for stratifying the severity of in human dengue disease. The design of the experiments is largely adequate and has yielded data are mostly supportive of the conclusions drawn here.

Major issues that affect impact:

– The improvements in disease symptoms appeared to be independent of the viral load. Since AC2-26 had previously been shown to have anti-inflammatory properties these findings are, while intriguing in the context of disease, not necessarily novel.

– It is well established that DENV does not replicate efficiently in mice, but infection can occur using mouse-adapted strains of DENV and/or immunocompromised mouse strains. Here, the authors use a combination of wt (Balb/C and C57BL/6) and mouse mutant lines (including AG129 mice which have been shown to support robust infection with mouse adapted DENV strains) and infect them with a very high dose of DENV2 isolate. There are very limited viremia data shown throughout the manuscript which makes is somewhat difficult to interpret some of the results. How much are the effects in all the non-AG129 mouse lines attributable to an acute response to the high inoculum (and no/very limited replication) vs actual infection?

– In Figure 4, when using IFNα/βR-/- mice (AG129), IL-6 and CCL5 instead of MCPT-1 and CCL2 were used as markers of systemic inflammatory mediators, which is not consistent with the previous experiments done in immunocompetent mice. No results of MCPT-1 and CCL2 changes in AG129 mice was presented and the authors didn't explain why they changed the markers in this manuscript. Since MCPT-1 is a mast cell-specific product, if it is not decreased in AG129 mice treated with Ac2-26 after DENV infection, how would the authors explain the protective effects of Ac2-26 in AG129 mice in the context of DENV infection?

– The authors showed AnxA1 memetic peptide Ac2-26 could suppress MC degranulation. However, the same experiments should also be done in parallel in AnxA1-/- and FPR2/ALX-/- mice and in BMMCs derived from them. If Ac2-26 can prevent MC degranulation, it is expected to see decreased degranulation in AnxA1 mice but not FPR2/ALX KO mice when infected with DENV, both in vivo and in BMMCs.

*Reviewer #3 (Recommendations for the authors):*

Costa and colleagues investigated the involvement of the pro-resolving protein Annexin A1 (AnxA1) in controlling vascular leakage and cytokine storm associated with dengue disease most severe forms. They find a decrease of AnxA1 levels in the plasma of patients during acute dengue virus infection compared to healthy patients. This decrease is even more pronounced in patients with severe dengue, i.e. requiring hospitalization, compared to those with non-severe dengue. A similar AnxA1 profile was measured over time in dengue-infected wild-type mice of two different strains and further investigated in this model. Animals lacking AnxA1 or AnxA1-receptor FPR2/ALX showed more severe pathology than wild-type mice, while maintaining unchanged viral load. Importantly, the administration of an AnxA1 mimetic peptide strongly attenuated the manifestations of dengue disease in wild-type and AnxA1 knockout mice. This effect could be explained in part by the limitation of mast cell degranulation in vivo and in vitro.

The strength of the article lies in the combination of observations made in patients with reproducible phenotypes in different mouse models, as well as in the testing of molecules with therapeutic potential. The manuscript is scientifically sound and the experiments are well executed. In general, the data are of very good quality, well analyzed and clearly presented. This study provides a very interesting perspective on the pharmacology of resolution, highlighting the benefit of expanding the scope of therapies in the future beyond that of viral load control in order to limit the overshoot of immune responses and associated damage.

---

## [Author Response]

Essential revisions:1. The improvements in disease symptoms appeared to be independent of the viral load. Since AC2-26 had previously been shown to have anti-inflammatory properties these findings are, while intriguing in the context of disease, not necessarily novel.

We have addressed this valid point in our reply to Reviewer #2 (comment #1). The relevance and novelty of our findings were further highlighted in the updated version of our manuscript.

The authors should mention that corticosteroids have been previously examined for the treatment of dengue disease and may exert beneficial effects through the induction of Annexin A1 and ALXR expression. They should also discuss why FPR2/ALX agonists may be better than treatment with corticosteroids to suppress exaggerated inflammation.

We have addressed this important point in our reply to Reviewer #1 (comment #1). Following the Reviewer's advice, the revised manuscript includes a discussion highlighting (i) the potential involvement of AnxA1 in the beneficial actions of corticosteroids in dengue disease and (ii) the therapeutic benefit of targeting this pathway over established anti-inflammatory therapies.

2. It is well established that DENV does not replicate efficiently in mice, but infection can occur using mouse-adapted strains of DENV and/or immunocompromised mouse strains. Here, the authors use a combination of wt (Balb/C and C57BL/6) and mouse mutant lines (including AG129 mice which have been shown to support robust infection with mouse adapted DENV strains) and infect them with a very high dose of DENV2 isolate. There are very limited viremia data shown throughout the manuscript which makes is somewhat difficult to interpret some of the results. How much are the effects in all the non-AG129 mouse lines attributable to an acute response to the high inoculum (and no/very limited replication) vs actual infection?

We have carefully addressed this concern in our reply to Reviewer #2 (comment #2) and included a new paragraph in the revised manuscript discussing this important point.

3. In Figure 4, when using IFNα/βR-/- mice (AG129), IL-6 and CCL5 instead of MCPT-1 and CCL2 were used as markers of systemic inflammatory mediators, which is not consistent with the previous experiments done in immunocompetent mice. No results of MCPT-1 and CCL2 changes in AG129 mice was presented and the authors didn't explain why they changed the markers in this manuscript. Since MCPT-1 is a mast cell-specific product, if it is not decreased in AG129 mice treated with Ac2-26 after DENV infection, how would the authors explain the protective effects of Ac2-26 in AG129 mice in the context of DENV infection?

As suggested by the Reviewer, we have included data on the systemic levels of MCPT-1 and CCL2 in A129 mice subjected to the model of severe dengue (Figure 4H,I). Please, refer to our reply to Reviewer #2 (comment #3).

4. The authors showed AnxA1 memetic peptide Ac2-26 could suppress MC degranulation. However, the same experiments should also be done in parallel in AnxA1-/- and FPR2/ALX-/- mice and in BMMCs derived from them. If Ac2-26 can prevent MC degranulation, it is expected to see decreased degranulation in AnxA1 mice but not FPR2/ALX KO mice when infected with DENV, both in vivo and in BMMCs.

As suggested by the Reviewer, we include new data in the revised version of the manuscript showing the dependence of FPR2/ALX for the modulatory effect of Ac_2-26_ in mast cell degranulation (Figure 6B,C).

Finally, the peptide concentrations in mice (150 µg/animal) and in vitro (100 µM) are pretty high. The authors should mention whether any side effects were observed.

As detailed in our reply to Reviewer #1 (comment #5), the in vivo dose used in our study was based on previous publications from our group and others (Damazo, Yona, Flower, Perretti, and Oliani, 2006; Galvao et al., 2017; Machado et al., 2020; Perretti, Ahluwalia, Harris, Goulding, and Flower, 1993; Vago et al., 2012). Using 150 µg/animal, we did not observe any notable adverse effects in our experiments (we have included this observation in the revised text).

Our in vitro concentration of Ac_2-26_ is consistent with previous studies on mast cells, where the prevention of mast cell degranulation is only achieved with high peptide concentrations (M. P. Oliveira, Prates, Gimenes, Correa, and Oliani, 2021). Besides, the concentration of Ac_2-26_ used in our in vitro studies was based on a concentration-response performed in bone-marrow-derived mast cells (Figure 6D of the updated manuscript).

Reviewer #1 (Recommendations for the authors):1. The authors should mention that corticosteroids have been previously examined for the treatment of dengue disease and may exert beneficial effects through the induction of Annexin A1 and ALXR expression. They should also discuss why FPR2/ALX agonists may be better than treatment with corticosteroids to suppress exaggerated inflammation.

We thank the Reviewer for their insightful comment. Although guidelines from the World Health Organization for managing dengue fever only provide recommendations on fluid management and supportive care (WHO, ‎2009), the use of anti-inflammatories in dengue disease has been evaluated. While the use of non-steroidal anti-inflammatory agents in dengue is discouraged, as they can increase the risk of bleeding (PAHO, 2016; WHO, ‎2009), recent studies have pointed towards a benefit of using corticoids in severe dengue with no significant adverse consequences or promotion of virus replication (S. M. R. Bandara and Herath, 2018; S. M. Rathnasiri Bandara and Herath, 2020). Noteworthy, amongst the several genomic and non-genomic actions of glucocorticoids, promotion of AnxA1 has shown to be relevant to the benefits afforded by steroids (De Caterina et al., 1993; Solito et al., 2003), as suggested by the loss of the therapeutic effect of glucocorticoids on AnxA1 depleted animals (Damazo et al., 2006; Patel et al., 2012; Perretti and D'Acquisto, 2009; Vago et al., 2012).

Our study reveals a central role of the axis AnxA1-FPR2/ALX in the pathogenesis of dengue disease. We suggest that this critical anti-inflammatory/pro-resolving pathway is downregulated during dengue, contributing to unbalanced inflammation and the cytokine storm characteristic of the disease. Therefore, there is scope to hypothesise that the benefit of corticosteroids in dengue disease could be partially attributed to promoting the glucocorticoid-inducible Annexin A1/FPR2/ALX pathway. Further investigations should confirm AnxA1 as an underlying mechanism of corticosteroids in dengue disease.

The effectiveness of the treatment with AnxA1 peptide in our study suggests that glucocorticoid therapy could be refined using a more targeted approach, particularly in infectious diseases where there is an underlying imbalance in the AnxA1 pathway. This opens an opportunity to explore small molecules targeting FPR2/ALX receptors, avoiding the spanning adverse consequences of glucocorticoids while targeting the defect in inflammation resolution operating in the disease settings. Further investigation on GC and GC-inducible AnxA1 is particularly relevant in dengue disease, where non-steroidal anti-inflammatory agents are discouraged, as mentioned above.

The revised manuscript includes a discussion highlighting (i) the potential involvement of AnxA1 in the beneficial actions of corticosteroids in dengue disease and (ii) the therapeutic benefit of targeting this pathway over established anti-inflammatory therapies (lines 620-640; 645).

2. Some references seem flawed (e.g. line 53, The Lancet Infectious, 2018). Should be checked throughout.

Thank you for noting this. We have replaced the reference in line 53 and thoroughly revised the other references in the manuscript.

3. Line 254: "Groups were comparable for sex distribution". Table 1 shows that the proportion of females was higher in the control group and that, on average, age was higher in patients with severe dengue disease. This should be mentioned in the text.

Thank you for your comment. Higher age in severe dengue cases is expected since older age is a risk factor for progression to severe disease (Rowe et al., 2014; Sangkaew et al., 2021). We have highlighted this age distribution in the updated version of our manuscript (line 263). As depicted in Table 1 legend, the sex ratio was statistically similar between the groups (p>0.1), making the groups comparable for sex distribution.

4. It would be interesting to know whether the levels of Annexin A1 in individuals with dengue infection (Figure 1A) are inversely correlated with the levels of inflammatory cytokines.

Although we agree with the Reviewer that this could be an interesting point to be addressed, we do not hold additional plasma samples to perform further ELISA assays. However, our research group is conducting an in-depth analysis of imbalances between inflammatory and pro-resolving mediators in plasma samples from dengue patients that will be comprehensively covered in a future publication.

Of note, in both models of dengue disease (immunocompetent and IFNα/βR KO animals), we show that decreased levels of AnxA1 during DENV infection are mirrored by higher levels of inflammatory markers, including the mast cell-specific product.

5. The peptide concentrations in mice (150 µg/animal) and in vitro (100 µM) are pretty high. The authors should mention whether any side effects were observed.

The dose of Ac_2-26_ peptide (150μg/animal) was based on previous publications, such as those in the settings of peritonitis (Damazo et al., 2006), gout (Galvao et al., 2017), air pouch (Perretti et al., 1993), pleurisy (Vago et al., 2012) and lung infection (Machado et al., 2020). Initial studies on Ac_2-26_ have used a dose of 200 µg per animal to prevent leukocyte migration induced by IL-1β, with no reports of adverse effects (Perretti et al., 1993). In addition, we did not observe any notable adverse effect in our experiments, as mentioned in line 390 of the updated manuscript.

Our in vitro concentration of Ac_2-26_ is consistent with previous studies on mast cells (cell lines, BMMC and subcutaneous tissue fragments), where prevention of mast cell degranulation is only achieved with a concentration in the order of micrograms per millilitre (M. P. Oliveira et al., 2021). To confirm the requirement of a high concentration of Ac_2-26_ in vitro, we performed a concentration-response, as depicted in Figure 6D of the updated manuscript.

6. The Annexin A1 levels in patients were about 10-fold lower than in mice (Figure 1). This difference should be discussed.

We thank the Reviewer for their comment. The plasma levels observed in our study are in line with previous measurements performed with a variety of studies using homemade ELISA assays or commercial ELISA kits (Adel et al., 2020; Purvis et al., 2019; Santana et al., 2021). Significant differences between animal and human samples in the same study were noticed before (Senchenkova et al., 2019; Xu et al., 2021). We have added this discussion in line 284 of the manuscript.

7. The authors mention that treatment with the Ac2-26 peptide did not affect systemic viral burden. It seems somewhat surprising that attenuation of innate immune activation doesn't affect the levels of viral replication. Please discuss.

This is an important point that has been discussed throughout the manuscript and further highlighted in the revised document (lines 89; 623; 648).

The ability of pro-resolving drugs to balance inflammation without hampering the host's ability to deal with pathogens is, in fact, a hallmark of resolution pharmacology, representing an advantage compared to standard anti-inflammatory drugs (Perretti, Leroy, Bland, and Montero-Melendez, 2015).

Several studies have suggested that pro-resolving mediators can suppress inflammation and diminish inflammatory responses while improving the phagocytosis of pathogens. This was initially demonstrated for specialised lipid mediators in bacterial infection (Chiang et al., 2012; Decker, Sadhu, and Fredman, 2021; Sekheri, El Kebir, Edner, and Filep, 2020), with subsequent studies with Annexin A1 on various infection models (Gobbetti et al., 2014; Machado et al., 2020; L. G. Oliveira et al., 2017; Tzelepis et al., 2015; Vanessa et al., 2015). Therefore, it is already well accepted that pro-resolving drugs can promote innate immunity against bacterial pathogens while reducing the deleterious effects of inflammation. Here we translate these findings into a model of infection. Please refer to our reply to Reviewer #2 (comment 1), where we discuss the novelty of our study.

Reviewer #2 (Recommendations for the authors):Major issues that affect impact:– The improvements in disease symptoms appeared to be independent of the viral load. Since AC2-26 had previously been shown to have anti-inflammatory properties these findings are, while intriguing in the context of disease, not necessarily novel.

Thank you for your comment. As discussed through the manuscript, the protective effect of AnxA1 and its mimetic peptide have been mainly demonstrated in sterile settings (e.g. inflammation triggered by LPS, uric acid crystals, DSS, and atherosclerosis genetically engineered animals), revealing various anti-inflammatory and pro-resolving actions of these molecules, including the promotion of granulocyte apoptosis, efferocytosis, and macrophage reprogramming (Fredman et al., 2015; Galvao et al., 2017; Gimenes et al., 2015; Kusters et al., 2015; Leoni et al., 2015; Locatelli et al., 2014; Sugimoto, Vago, Perretti, and Teixeira, 2019). Later, studies suggested that pro-resolving mediators can suppress inflammation and reduce the inflammatory response while improving the phagocytosis of pathogens. This was initially demonstrated for specialised lipid mediators in bacterial infection (Chiang et al., 2012; Decker et al., 2021; Sekheri et al., 2020), with subsequent studies with Annexin A1 on various infection models (Gobbetti et al., 2014; Machado et al., 2020; L. G. Oliveira et al., 2017; Tzelepis et al., 2015; Vanessa et al., 2015).

However, data on the role of resolving mediators on viral infections are scarce, with most current studies focusing on lipidic mediators of resolution (Walker et al., 2021) and insufficient data on the role of Annexin A1 in viral diseases (Schloer et al., 2019). Findings that Annexin A1 could promote influenza A virus replication in vitro were particularly surprising (Arora et al., 2016). Therefore, while studies on bacterial infection consistently revealed the ability of pro-resolving drugs to facilitate innate immune responses against the pathogen while reducing the deleterious effects of inflammation, the translational potential of these results to viral infection was less clear.

Our work is the first to access the protective role of AnxA1 in dengue infection and to reveal the therapeutic potential of Annexin A1 mimetic peptides. We show that AnxA1 can suppress the inflammatory response without hampering the protective immunity against the virus. Besides, although previous studies have suggested that AnxA1 could suppress mast cell degranulation (Gimenes et al., 2015; M. P. Oliveira et al., 2021; Parisi, Correa, and Gil, 2019; Stuqui et al., 2015), our manuscript is the first to describe the role of AnxA1 on mast cells degranulation triggered by DENV.

Finally, our findings are relevant since pathogenic immune responses are not exclusive to dengue but underlie the severity of several other viral diseases, including severe community-acquired pneumonia caused by viruses (D'Elia, Harrison, Oyston, Lukaszewski, and Clark, 2013; Perrone, Plowden, Garcia-Sastre, Katz, and Tumpey, 2008) and COVID-19 (Jose and Manuel, 2020). Therefore, other researchers could explore our data and translate it into different infectious settings, whereby targeting the AnxA1 pathway, with or without combination with antiviral drugs, holds promising therapeutic potential.

The relevance and novelty of our findings were further highlighted in the updated version of our manuscript (lines 89; 623; 648).

– It is well established that DENV does not replicate efficiently in mice, but infection can occur using mouse-adapted strains of DENV and/or immunocompromised mouse strains. Here, the authors use a combination of wt (Balb/C and C57BL/6) and mouse mutant lines (including AG129 mice which have been shown to support robust infection with mouse adapted DENV strains) and infect them with a very high dose of DENV2 isolate. There are very limited viremia data shown throughout the manuscript which makes is somewhat difficult to interpret some of the results. How much are the effects in all the non-AG129 mouse lines attributable to an acute response to the high inoculum (and no/very limited replication) vs actual infection?

Despite the limitations of modelling dengue disease in immunocompetent animals, in our study, AnxA1 and FPR2/ALX knockout animals (and their respective littermates BALB/c and C57BL/6 mice) were used as a genetic tool to investigate the role of endogenous AnxA1 in dengue disease. As mentioned by the Reviewer, DENV low passage clinical isolates do not replicate efficiently in immunocompetent mice, and for that reason, the inoculum must be higher than when using adapted strains to induce a similar disease profile. Of note, here we used 1x10^6^ PFU of DENV-2 per mouse (for immunocompetent strains): this is the same virus strain and inoculum used in recent publications (Rathore et al., 2019; St John, Rathore, Raghavan, Ng, and Abraham, 2013). Although viremia is not detectable in immunocompetent mice (St John et al., 2013), we noticed transient viral RNA detection in the spleen starting 24h after infection, peaking at 72h, followed by disappearance at 96h (Figure 2_supplement 1 and Figure 3_supplement 1). These data are in line with those presented in previous studies using the same model (St John et al., 2013; Syenina, Jagaraj, Aman, Sridharan, and St John, 2015).

Although resistant to DENV infection (Shresta et al., 2004; Yauch and Shresta, 2008), it has been reported that infection of immunocompetent mice with DENV can induce a self-limited disease, like dengue fever. While limited by the lack of detectable viremia, these models allow the study of critical immune components, such as mast cells (Rathore et al., 2019; St John et al., 2013; Syenina et al., 2015), and their contribution to the classical haematological abnormalities and altered vascular permeability observed during DENV infection (Assuncao-Miranda et al., 2010; Chen, Hofman, Kung, Lin, and Wu-Hsieh, 2007).

To complement our studies with Balb/c and C57Bl/6 mice, we have performed experiments in A129 mice since they are known to be highly susceptible to flaviviruses infections (de Freitas et al., 2019; Del Sarto et al., 2020) and develop a more severe disease manifestation that emulates several features of severe dengue in humans (Zandi et al., 2019). This model is considered a gold standard viremia model to test potential antiviral therapies (Meyts and Casanova, 2021; Williams, Zompi, Beatty, and Harris, 2009). Using this model, we aimed to evaluate if the treatment with peptide Ac_2-26_ impacts viral loads, besides the alterations on disease and inflammatory parameters observed in immunocompetent animals. We have successfully recovered viable virus from samples using the plaque assay technique, as described in the Method section of the manuscript. Since A129 animals are more susceptible to infection, we could use a lower inoculum of the same non-adapted strain in this model (10^3^ PFU).

Therefore, we have applied two different techniques (quantification of viral RNA for immunocompetent mice and quantification of viable particles for A129 KO mice) and two different models (high and low DENV inoculums) to investigate the impact of endogenous and exogenous AnxA1 on the host ability to deal with the infection.

We have included a new paragraph in the revised manuscript discussing this important point (lines 586-605).

– In Figure 4, when using IFNα/βR-/- mice (AG129), IL-6 and CCL5 instead of MCPT-1 and CCL2 were used as markers of systemic inflammatory mediators, which is not consistent with the previous experiments done in immunocompetent mice. No results of MCPT-1 and CCL2 changes in AG129 mice was presented and the authors didn't explain why they changed the markers in this manuscript. Since MCPT-1 is a mast cell-specific product, if it is not decreased in AG129 mice treated with Ac2-26 after DENV infection, how would the authors explain the protective effects of Ac2-26 in AG129 mice in the context of DENV infection?

Thank you for this important comment. The inflammatory markers investigated in the model of severe dengue (A129 animals) were selected based on previous publications showing CCL5 and IL6 as classical inflammatory markers in dengue and a requirement of CCR5 for DENV replication and infection development (Guabiraba et al., 2010; Marques et al., 2015). Besides, liver injury, a feature of dengue disease observed in our A129 model (Binh, Matheus, Huong, Deparis, and Marechal, 2009; Trung et al., 2010) seems to correlate with high levels of CCL5 (Conceicao et al., 2010; Suksanpaisan, Cabrera-Hernandez, and Smith, 2007),

As suggested by the Reviewer, we have included data on the systemic levels of MCPT-1 and CCL2 in A129 mice subjected to the model of severe dengue (Figure 4H-J; line 383). As observed in immunocompetent mice, Ac_2-26_ attenuated the systemic release of the mast cell chymase MCPT-1 induced by DENV-2 (Figure 4H), although it did not affect systemic levels of CCL2 (Figure 4I). The lack of effect of Ac_2-26_ in CCL2 levels might be justified by the higher severity of the model in A129 animals, resulting in significantly higher production of CCL2 compared to immunocompetent animals infected with DENV (mean ± SD CCL2 production at the peak of 585.0 ± 61.9 for C57BL/6; 621.2 ± 100.5 for BALB/c; and 977.8 ± 110.7 for A129 mice).

– The authors showed AnxA1 memetic peptide Ac2-26 could suppress MC degranulation. However, the same experiments should also be done in parallel in AnxA1-/- and FPR2/ALX-/- mice and in BMMCs derived from them. If Ac2-26 can prevent MC degranulation, it is expected to see decreased degranulation in AnxA1 mice but not FPR2/ALX KO mice when infected with DENV, both in vivo and in BMMCs.

Thank you for your suggestion. We have included new data in the revised version of the manuscript showing the dependence of FPR2/ALX for the modulatory effect of Ac_2-26_ in mast cell degranulation (Figure 6B,C; methods lines 201-207; results lines 461-469).

As predicted by the Reviewer, we verified that systemic administration of Ac_2-26_ successfully reduced mast cell degranulation in AnxA1 KO mice (Figure 6B), but failed to rescue DENV-2 induced degranulation in FPR2/ALX KO animals (Figure 6C). Notably, the absence of FPR2/ALX significantly increased mast cell degranulation in response to the local challenge with DENV compared to the WT counterparts (Figure 6C).

References

Adel, F. W., Rikhi, A., Wan, S. H., Iyer, S. R., Chakraborty, H., McNulty, S.,... Chen, H. H. (2020). Annexin A1 is a Potential Novel Biomarker of Congestion in Acute Heart Failure. *J Card Fail, 26*(8), 727-732. doi:10.1016/j.cardfail.2020.05.012

Arora, S., Lim, W., Bist, P., Perumalsamy, R., Lukman, H. M., Li, F.,... Lim, L. H. (2016). Influenza A virus enhances its propagation through the modulation of Annexin-A1 dependent endosomal trafficking and apoptosis. *Cell Death Differ, 23*(7), 1243-1256. doi:10.1038/cdd.2016.19

Assuncao-Miranda, I., Amaral, F. A., Bozza, F. A., Fagundes, C. T., Sousa, L. P., Souza, D. G.,... Bozza, M. T. (2010). Contribution of macrophage migration inhibitory factor to the pathogenesis of dengue virus infection. *FASEB J, 24*(1), 218-228. doi:10.1096/fj.09-139469

Bandara, S. M. R., and Herath, H. (2018). Effectiveness of corticosteroid in the treatment of dengue – A systemic review. *Heliyon, 4*(9), e00816. doi:10.1016/j.heliyon.2018.e00816

Bandara, S. M. R., and Herath, H. M. M. T. B. (2020). Corticosteroid actions on dengue immune pathology; A review article. *Clinical Epidemiology and Global Health, 8*(2), 486-494. doi:10.1016/j.cegh.2019.11.001

Binh, P. T., Matheus, S., Huong, V. T., Deparis, X., and Marechal, V. (2009). Early clinical and biological features of severe clinical manifestations of dengue in Vietnamese adults. *J Clin Virol, 45*(4), 276-280. doi:10.1016/j.jcv.2009.04.004

Chen, H. C., Hofman, F. M., Kung, J. T., Lin, Y. D., and Wu-Hsieh, B. A. (2007). Both virus and tumor necrosis factor α are critical for endothelium damage in a mouse model of dengue virus-induced hemorrhage. *J Virol, 81*(11), 5518-5526. doi:10.1128/JVI.02575-06

Chiang, N., Fredman, G., Backhed, F., Oh, S. F., Vickery, T., Schmidt, B. A., and Serhan, C. N. (2012). Infection regulates pro-resolving mediators that lower antibiotic requirements. *Nature, 484*(7395), 524-528. doi:10.1038/nature11042

Conceicao, T. M., El-Bacha, T., Villas-Boas, C. S., Coello, G., Ramirez, J., Montero-Lomeli, M., and Da Poian, A. T. (2010). Gene expression analysis during dengue virus infection in HepG2 cells reveals virus control of innate immune response. *J Infect, 60*(1), 65-75. doi:10.1016/j.jinf.2009.10.003

D'Elia, R. V., Harrison, K., Oyston, P. C., Lukaszewski, R. A., and Clark, G. C. (2013). Targeting the "cytokine storm" for therapeutic benefit. *Clin Vaccine Immunol, 20*(3), 319-327. doi:10.1128/CVI.00636-12

Damazo, A. S., Yona, S., Flower, R. J., Perretti, M., and Oliani, S. M. (2006). Spatial and temporal profiles for anti-inflammatory gene expression in leukocytes during a resolving model of peritonitis. *J Immunol, 176*(7), 4410-4418. doi:10.4049/jimmunol.176.7.4410

De Caterina, R., Sicari, R., Giannessi, D., Paggiaro, P. L., Paoletti, P., Lazzerini, G.,... Parente, L. (1993). Macrophage-specific eicosanoid synthesis inhibition and lipocortin-1 induction by glucocorticoids. *J Appl Physiol (1985), 75*(6), 2368-2375. doi:10.1152/jappl.1993.75.6.2368

de Freitas, C. S., Higa, L. M., Sacramento, C. Q., Ferreira, A. C., Reis, P. A., Delvecchio, R.,... Souza, T. M. L. (2019). Yellow fever virus is susceptible to sofosbuvir both in vitro and in vivo. *PLoS Negl Trop Dis, 13*(1), e0007072. doi:10.1371/journal.pntd.0007072

Decker, C., Sadhu, S., and Fredman, G. (2021). Pro-Resolving Ligands Orchestrate Phagocytosis. *Front Immunol, 12*, 660865. doi:10.3389/fimmu.2021.660865

Del Sarto, J. L., de Paiva Froes Rocha, R., Bassit, L., Olmo, I. G., Valiate, B., Queiroz-Junior, C. M.,... Teixeira, M. M. (2020). 7-Deaza-7-Fluoro-2'-C-Methyladenosine Inhibits Zika virus Infection and Viral-induced Neuroinflammation. *Antiviral Res*, 104855. doi:10.1016/j.antiviral.2020.104855

Fredman, G., Kamaly, N., Spolitu, S., Milton, J., Ghorpade, D., Chiasson, R.,... Tabas, I. (2015). Targeted nanoparticles containing the proresolving peptide Ac2-26 protect against advanced atherosclerosis in hypercholesterolemic mice. *Sci Transl Med, 7*(275), 275ra220. doi:10.1126/scitranslmed.aaa1065

Galvao, I., Vago, J. P., Barroso, L. C., Tavares, L. P., Queiroz-Junior, C. M., Costa, V. V.,... Teixeira, M. M. (2017). Annexin A1 promotes timely resolution of inflammation in murine gout. *Eur J Immunol, 47*(3), 585-596. doi:10.1002/eji.201646551

Gimenes, A. D., Andrade, T. R., Mello, C. B., Ramos, L., Gil, C. D., and Oliani, S. M. (2015). Beneficial effect of annexin A1 in a model of experimental allergic conjunctivitis. *Exp Eye Res, 134*, 24-32. doi:10.1016/j.exer.2015.03.013

Gobbetti, T., Coldewey, S. M., Chen, J., McArthur, S., le Faouder, P., Cenac, N.,... Perretti, M. (2014). Nonredundant protective properties of FPR2/ALX in polymicrobial murine sepsis. *Proc Natl Acad Sci U S A, 111*(52), 18685-18690. doi:10.1073/pnas.1410938111

Guabiraba, R., Marques, R. E., Besnard, A. G., Fagundes, C. T., Souza, D. G., Ryffel, B., and Teixeira, M. M. (2010). Role of the chemokine receptors CCR1, CCR2 and CCR4 in the pathogenesis of experimental dengue infection in mice. *PLoS ONE, 5*(12), e15680. doi:10.1371/journal.pone.0015680

Jose, R. J., and Manuel, A. (2020). COVID-19 cytokine storm: the interplay between inflammation and coagulation. *The Lancet Respiratory Medicine, 8*(6), e46-e47. doi:10.1016/s2213-2600(20)30216-2

Kusters, D. H., Chatrou, M. L., Willems, B. A., De Saint-Hubert, M., Bauwens, M., van der Vorst, E.,... Reutelingsperger, C. P. (2015). Pharmacological Treatment with Annexin A1 Reduces Atherosclerotic Plaque Burden in LDLR-/- Mice on Western Type Diet. *PLoS ONE, 10*(6), e0130484. doi:10.1371/journal.pone.0130484

Leoni, G., Neumann, P. A., Kamaly, N., Quiros, M., Nishio, H., Jones, H. R.,... Nusrat, A. (2015). Annexin A1-containing extracellular vesicles and polymeric nanoparticles promote epithelial wound repair. *J Clin Invest, 125*(3), 1215-1227. doi:10.1172/JCI76693

Locatelli, I., Sutti, S., Jindal, A., Vacchiano, M., Bozzola, C., Reutelingsperger, C.,... Perretti, M. (2014). Endogenous annexin A1 is a novel protective determinant in nonalcoholic steatohepatitis in mice. *Hepatology, 60*(2), 531-544. doi:10.1002/*hep*.27141

Machado, M. G., Tavares, L. P., Souza, G. V. S., Queiroz-Junior, C. M., Ascencao, F. R., Lopes, M. E.,... Sousa, L. P. (2020). The Annexin A1/FPR2 pathway controls the inflammatory response and bacterial dissemination in experimental pneumococcal pneumonia. *FASEB J, 34*(2), 2749-2764. doi:10.1096/fj.201902172R

Marques, R. E., Guabiraba, R., Del Sarto, J. L., Rocha, R. F., Queiroz, A. L., Cisalpino, D.,... Teixeira, M. M. (2015). Dengue virus requires the CC-chemokine receptor CCR5 for replication and infection development. *Immunology, 145*(4), 583-596. doi:10.1111/imm.12476

Meyts, I., and Casanova, J. L. (2021). Viral infections in humans and mice with genetic deficiencies of the type I IFN response pathway. *Eur J Immunol, 51*(5), 1039-1061. doi:10.1002/eji.202048793

Oliveira, L. G., Souza-Testasicca, M. C., Vago, J. P., Figueiredo, A. B., Canavaci, A. M., Perucci, L. O.,... Fernandes, A. P. (2017). Annexin A1 Is Involved in the Resolution of Inflammatory Responses during Leishmania braziliensis Infection. *J Immunol, 198*(8), 3227-3236. doi:10.4049/jimmunol.1602028

Oliveira, M. P., Prates, J., Gimenes, A. D., Correa, S. G., and Oliani, S. M. (2021). Annexin A1 Mimetic Peptide Ac2-26 Modulates the Function of Murine Colonic and Human Mast Cells. *Front Immunol, 12*, 689484. doi:10.3389/fimmu.2021.689484

PAHO. (2016). *Dengue: guidelines for patient care in the Region of the Americas..*

Parisi, J. D. S., Correa, M. P., and Gil, C. D. (2019). Lack of Endogenous Annexin A1 Increases Mast Cell Activation and Exacerbates Experimental Atopic Dermatitis. *Cells, 8*(1). doi:10.3390/cells8010051

Patel, H. B., Kornerup, K. N., Sampaio, A. L., D'Acquisto, F., Seed, M. P., Girol, A. P.,... Perretti, M. (2012). The impact of endogenous annexin A1 on glucocorticoid control of inflammatory arthritis. *Ann Rheum Dis, 71*(11), 1872-1880. doi:10.1136/annrheumdis-2011-201180

Perretti, M., Ahluwalia, A., Harris, J. G., Goulding, N. J., and Flower, R. J. (1993). Lipocortin-1 fragments inhibit neutrophil accumulation and neutrophil-dependent edema in the mouse. A qualitative comparison with an anti-CD11b monoclonal antibody. *J Immunol, 151*(8), 4306-4314.

Perretti, M., and D'Acquisto, F. (2009). Annexin A1 and glucocorticoids as effectors of the resolution of inflammation. *Nat Rev Immunol, 9*(1), 62-70. doi:10.1038/nri2470

Perretti, M., Leroy, X., Bland, E. J., and Montero-Melendez, T. (2015). Resolution Pharmacology: Opportunities for Therapeutic Innovation in Inflammation. *Trends Pharmacol Sci, 36*(11), 737-755. doi:10.1016/j.tips.2015.07.007

Perrone, L. A., Plowden, J. K., Garcia-Sastre, A., Katz, J. M., and Tumpey, T. M. (2008). H5N1 and 1918 pandemic influenza virus infection results in early and excessive infiltration of macrophages and neutrophils in the lungs of mice. *PLoS Pathog, 4*(8), e1000115. doi:10.1371/journal.ppat.1000115

Purvis, G. S. D., Collino, M., Loiola, R. A., Baragetti, A., Chiazza, F., Brovelli, M.,... Thiemermann, C. (2019). Identification of AnnexinA1 as an Endogenous Regulator of RhoA, and Its Role in the Pathophysiology and Experimental Therapy of Type-2 Diabetes. *Front Immunol, 10*, 571. doi:10.3389/fimmu.2019.00571

Rathore, A. P., Mantri, C. K., Aman, S. A., Syenina, A., Ooi, J., Jagaraj, C. J.,... St John, A. L. (2019). Dengue virus-elicited tryptase induces endothelial permeability and shock. *J Clin Invest, 129*(10), 4180-4193. doi:10.1172/JCI128426

Rowe, E. K., Leo, Y. S., Wong, J. G., Thein, T. L., Gan, V. C., Lee, L. K., and Lye, D. C. (2014). Challenges in dengue fever in the elderly: atypical presentation and risk of severe dengue and hospital-acquired infection [corrected]. *PLoS Negl Trop Dis, 8*(4), e2777. doi:10.1371/journal.pntd.0002777

Sangkaew, S., Ming, D., Boonyasiri, A., Honeyford, K., Kalayanarooj, S., Yacoub, S.,... Holmes, A. (2021). Risk predictors of progression to severe disease during the febrile phase of dengue: a systematic review and meta-analysis. *The Lancet Infectious Diseases, 21*(7), 1014-1026. doi:10.1016/s1473-3099(20)30601-0

Santana, B. B., Queiroz, M. A. F., Cerveira, R. A., Rodrigues, C. M., da Silva Graca Amoras, E., da Costa, C. A.,... Vallinoto, A. C. R. (2021). Low Annexin A1 level in HTLV-1 infected patients is a potential biomarker for the clinical progression and diagnosis of HAM/TSP. *BMC Infect Dis, 21*(1), 219. doi:10.1186/s12879-021-05917-y

Schloer, S., Hubel, N., Masemann, D., Pajonczyk, D., Brunotte, L., Ehrhardt, C.,... Rescher, U. (2019). The annexin A1/FPR2 signaling axis expands alveolar macrophages, limits viral replication, and attenuates pathogenesis in the murine influenza A virus infection model. *FASEB J*, fj201901265R. doi:10.1096/fj.201901265R

Sekheri, M., El Kebir, D., Edner, N., and Filep, J. G. (2020). 15-Epi-LXA4 and 17-epi-RvD1 restore TLR9-mediated impaired neutrophil phagocytosis and accelerate resolution of lung inflammation. *Proc Natl Acad Sci U S A, 117*(14), 7971-7980. doi:10.1073/pnas.1920193117

Senchenkova, E. Y., Ansari, J., Becker, F., Vital, S. A., Al-Yafeai, Z., Sparkenbaugh, E. M.,... Gavins, F. N. E. (2019). Novel Role for the AnxA1-Fpr2/ALX Signaling Axis as a Key Regulator of Platelet Function to Promote Resolution of Inflammation. *Circulation, 140*(4), 319-335. doi:10.1161/CIRCULATIONAHA.118.039345

Shresta, S., Kyle, J. L., Snider, H. M., Basavapatna, M., Beatty, P. R., and Harris, E. (2004). Interferon-dependent immunity is essential for resistance to primary dengue virus infection in mice, whereas T- and B-cell-dependent immunity are less critical. *J Virol, 78*(6), 2701-2710. doi:10.1128/jvi.78.6.2701-2710.2004

Solito, E., Mulla, A., Morris, J. F., Christian, H. C., Flower, R. J., and Buckingham, J. C. (2003). Dexamethasone induces rapid serine-phosphorylation and membrane translocation of annexin 1 in a human folliculostellate cell line via a novel nongenomic mechanism involving the glucocorticoid receptor, protein kinase C, phosphatidylinositol 3-kinase, and mitogen-activated protein kinase. *Endocrinology, 144*(4), 1164-1174. doi:10.1210/en.2002-220592

St John, A. L., Rathore, A. P., Raghavan, B., Ng, M. L., and Abraham, S. N. (2013). Contributions of mast cells and vasoactive products, leukotrienes and chymase, to dengue virus-induced vascular leakage. *eLife, 2*, e00481. doi:10.7554/*eLife*.00481

Stuqui, B., de Paula-Silva, M., Carlos, C. P., Ullah, A., Arni, R. K., Gil, C. D., and Oliani, S. M. (2015). Ac2-26 Mimetic Peptide of Annexin A1 Inhibits Local and Systemic Inflammatory Processes Induced by Bothrops moojeni Venom and the Lys-49 Phospholipase A2 in a Rat Model. *PLoS ONE, 10*(7), e0130803. doi:10.1371/journal.pone.0130803

Sugimoto, M. A., Vago, J. P., Perretti, M., and Teixeira, M. M. (2019). Mediators of the Resolution of the Inflammatory Response. *Trends Immunol, 40*(3), 212-227. doi:10.1016/j.it.2019.01.007

Suksanpaisan, L., Cabrera-Hernandez, A., and Smith, D. R. (2007). Infection of human primary hepatocytes with dengue virus serotype 2. *J Med Virol, 79*(3), 300-307. doi:10.1002/jmv.20798

Syenina, A., Jagaraj, C. J., Aman, S. A., Sridharan, A., and St John, A. L. (2015). Dengue vascular leakage is augmented by mast cell degranulation mediated by immunoglobulin Fcgamma receptors. *eLife, 4*. doi:10.7554/*eLife*.05291

Trung, D. T., Thao le, T. T., Hien, T. T., Hung, N. T., Vinh, N. N., Hien, P. T.,... Wills, B. (2010). Liver involvement associated with dengue infection in adults in Vietnam. *Am J Trop Med Hyg, 83*(4), 774-780. doi:10.4269/ajtmh.2010.10-0090

Tzelepis, F., Verway, M., Daoud, J., Gillard, J., Hassani-Ardakani, K., Dunn, J.,... Divangahi, M. (2015). Annexin1 regulates DC efferocytosis and cross-presentation during *Mycobacterium tuberculosis* infection. *J Clin Invest, 125*(2), 752-768. doi:10.1172/JCI77014

Vago, J. P., Nogueira, C. R., Tavares, L. P., Soriani, F. M., Lopes, F., Russo, R. C.,... Sousa, L. P. (2012). Annexin A1 modulates natural and glucocorticoid-induced resolution of inflammation by enhancing neutrophil apoptosis. *J Leukoc Biol, 92*(2), 249-258. doi:10.1189/jlb.0112008

Vanessa, K. H., Julia, M. G., Wenwei, L., Michelle, A. L., Zarina, Z. R., Lina, L. H., and Sylvie, A. (2015). Absence of Annexin A1 impairs host adaptive immunity against *Mycobacterium tuberculosis* in vivo. *Immunobiology, 220*(5), 614-623. doi:10.1016/j.imbio.2014.12.001

Walker, K. H., Krishnamoorthy, N., Bruggemann, T. R., Shay, A. E., Serhan, C. N., and Levy, B. D. (2021). Protectins PCTR1 and PD1 Reduce Viral Load and Lung Inflammation During Respiratory Syncytial Virus Infection in Mice. *Front Immunol, 12*, 704427. doi:10.3389/fimmu.2021.704427

WHO. (‎2009). *Dengue guidelines for diagnosis, treatment, prevention and control : new edition.* : World Health Organization.

Williams, K. L., Zompi, S., Beatty, P. R., and Harris, E. (2009). A mouse model for studying dengue virus pathogenesis and immune response. *Ann N Y Acad Sci, 1171 Suppl 1*, E12-23. doi:10.1111/j.1749-6632.2009.05057.x

Xu, X., Gao, W., Li, L., Hao, J., Yang, B., Wang, T.,... Jiao, L. (2021). Annexin A1 protects against cerebral ischemia-reperfusion injury by modulating microglia/macrophage polarisation via FPR2/ALX-dependent AMPK-mTOR pathway. *J Neuroinflammation, 18*(1), 119. doi:10.1186/s12974-021-02174-3

Yauch, L. E., and Shresta, S. (2008). Mouse models of dengue virus infection and disease. *Antiviral Res, 80*(2), 87-93. doi:10.1016/j.antiviral.2008.06.010

Zandi, K., Bassit, L., Amblard, F., Cox, B. D., Hassandarvish, P., Moghaddam, E.,... Schinazi, R. F. (2019). Nucleoside Analogs with Selective Antiviral Activity against Dengue Fever and Japanese Encephalitis Viruses. *Antimicrob Agents Chemother, 63*(7). doi:10.1128/AAC.00397-19